# Scalable Learning and MAP Inference for Nonsymmetric Determinantal Point Processes

**Mike Gartrell**
Criteo AI Lab
m.gartrell@criteo.com

**Insu Han**
KAIST
insu.han@kaist.ac.kr

**Elvis Dohmatob**
Criteo AI Lab
e.dohmatob@criteo.com

**Jennifer Gillenwater**
Google Research
jengi@google.com

**Victor-Emmanuel Brunel**
ENSAE ParisTech
victor.emmanuel.brunel@ensae.fr

## Abstract

Determinantal point processes (DPPs) have attracted significant attention in machine learning for their ability to model subsets drawn from a large item collection. Recent work shows that nonsymmetric DPP (NDPP) kernels have significant advantages over symmetric kernels in terms of modeling power and predictive performance. However, for an item collection of size $M$, existing NDPP learning and inference algorithms require memory quadratic in $M$ and runtime cubic (for learning) or quadratic (for inference) in $M$, making them impractical for many typical subset selection tasks. In this work, we develop a learning algorithm with space and time requirements linear in $M$ by introducing a new NDPP kernel decomposition. We also derive a linear-complexity NDPP maximum a posteriori (MAP) inference algorithm that applies not only to our new kernel but also to that of prior work. Through evaluation on real-world datasets, we show that our algorithms scale significantly better, and can match the predictive performance of prior work.

## 1 Introduction

Determinantal point processes (DPPs) have proven useful for numerous machine learning tasks. For example, recent uses include summarization (Sharghi et al., 2018), recommender systems (Wilhelm et al., 2018), neural network compression (Mariet & Sra, 2016), kernel approximation (Li et al., 2016), multi-modal output generation (Elfeki et al., 2019), and batch selection, both for stochastic optimization (Zhang et al., 2017) and for active learning (Bıyık et al., 2019). For subset selection problems where the ground set of items to select from has cardinality $M$, the typical DPP is parameterized by an $M \times M$ kernel matrix. Most prior work has been concerned with *symmetric* DPPs, where the kernel must equal its transpose. However, recent work has considered the more general class of *nonsymmetric* DPPs (NDPPs) and shown that these have additional useful modeling power (Brunel, 2018; Gartrell et al., 2019). In particular, unlike symmetric DPPs, which can only model negative correlations between items, NDPPs allow modeling of positive correlations, where the presence of item $i$ in the selected set increases the probability that some other item $j$ will also be selected. There are many intuitive examples of how positive correlations can be of practical importance. For example, consider a product recommendation task for a retail website, where a camera is found in a user's shopping cart, and the goal is to display several other items that might be purchased. Relative to an empty cart, the presence of the camera probably *increases* the probability of buying an accessory like a tripod.

Although NDPPs can theoretically model such behavior, the existing approach for NDPP learning and inference (Gartrell et al., 2019) is often impractical in terms of both storage and runtime requirements. These algorithms require memory quadratic in $M$ and time quadratic (for inference) or cubic (for learning) in $M$; for the not-unusual $M$ of 1 million, this requires storing 8TB-size objects in memory, with runtime millions or billions of times slower than that of a linear-complexity method.

In this work, we make the following contributions:

**Learning**: We propose a new decomposition of the NDPP kernel which reduces the storage and run-time requirements of learning and inference to linear in $M$. Fortuitously, the modified decomposition retains all of the previous decomposition's modeling power, as it covers the same part of the NDPP kernel space. The algebraic manipulations we apply to get linear complexity for this decomposition cannot be applied to prior work, meaning that our new decomposition is crucial for scalability.

**Inference**: After learning, prior NDPP work applies a DPP conditioning algorithm to do subset expansion (Gartrell et al., 2019), with quadratic runtime in $M$. However, prior work does not examine the general problem of MAP inference for NDPPs, i.e., solving the problem of finding the highest-probability subset under a DPP. For symmetric DPPs, there exists a standard greedy MAP inference algorithm that is linear in $M$. In this work, we develop a version of this algorithm that is also linear for low-rank NDPPs. The low-rank requirement is unique to NDPPs, and highlights the fact that the transformation of the algorithm from the symmetric to the nonsymmetric space is non-trivial. To the best of our knowledge, this is the first MAP algorithm proposed for NDPPs.

We combine the above contributions through experiments that involve learning NDPP kernels and applying MAP inference to these kernels to do subset selection for several real-world datasets. These experiments demonstrate that our algorithms are much more scalable, and that the new kernel decomposition matches the predictive performance of the decomposition from prior work.

## 2 BACKGROUND

Consider a finite set $\mathcal{Y} = \{1, 2, \dots, M\}$ of cardinality $M$, which we will also denote by $[\![M]\!]$. A DPP on $[\![M]\!]$ defines a probability distribution over all of its $2^M$ subsets. It is parameterized by a matrix $\boldsymbol{L} \in \mathbb{R}^{M \times M}$, called the *kernel*, such that the probability of each subset $Y \subseteq [\![M]\!]$ is proportional to the determinant of its corresponding principal submatrix: $\Pr(Y) \propto \det(\boldsymbol{L}_Y)$. The normalization constant for this distribution can be expressed as a single $M \times M$ determinant: $\sum_{Y \subseteq [\![M]\!]} \det(\boldsymbol{L}_Y) = \det(\boldsymbol{L} + \boldsymbol{I})$ (Kulesza et al., 2012, Theorem 2.1). Hence, $\Pr(Y) = \det(\boldsymbol{L}_Y) / \det(\boldsymbol{L} + \boldsymbol{I})$. We will use $\mathbb{P}_{\boldsymbol{L}}$ to denote this distribution.

For intuition about the kernel parameters, notice that the probabilities of singletons $\{i\}$ and $\{j\}$ are proportional to $\boldsymbol{L}_{ii}$ and $\boldsymbol{L}_{jj}$, respectively. Hence, it is common to think of $\boldsymbol{L}$'s diagonal as representing item qualities. The probability of a pair $\{i, j\}$ is proportional to $\det(\boldsymbol{L}_{\{i,j\}}) = \boldsymbol{L}_{ii}\boldsymbol{L}_{jj} - \boldsymbol{L}_{ij}\boldsymbol{L}_{ji}$. Thus, if $-\boldsymbol{L}_{ij}\boldsymbol{L}_{ji} < 0$, this indicates $i$ and $j$ interact negatively. Similarly, if $-\boldsymbol{L}_{ij}\boldsymbol{L}_{ji} > 0$, then $i$ and $j$ interact positively. Therefore, off-diagonal terms determine item interactions. (The vague term "interactions" can be replaced by the more precise term "correlations" if we consider the DPP's marginal kernel instead; see Gartrell et al. (2019, Section 2.1) for an extensive discussion.)

In order to ensure that $\mathbb{P}_{\boldsymbol{L}}$ defines a probability distribution, all principal minors of $\boldsymbol{L}$ must be non-negative: $\det(\boldsymbol{L}_Y) \geq 0$. Matrices that satisfy this property are called $P_0$-matrices (Fang, 1989, Definition 1). There is no known generative method or matrix decomposition that fully covers the space of all $P_0$ matrices, although there are many that partially cover the space (Tsatsomeros, 2004).

One common partial solution is to use a decomposition that covers the space of *symmetric* $P_0$ matrices. By restricting to the space of symmetric matrices, one can exploit the fact that $\boldsymbol{L} \in P_0$ if $\boldsymbol{L}$ is positive semidefinite (PSD)[*] (Prussing, 1986). Any symmetric PSD matrix can be written as the Gramian matrix of some set of vectors: $\boldsymbol{L} := \boldsymbol{V}\boldsymbol{V}^\top$, where $\boldsymbol{V} \in \mathbb{R}^{M \times K}$. Hence, the $\boldsymbol{V}\boldsymbol{V}^\top$ decomposition provides an easy means of generating the entire space of symmetric $P_0$ matrices. It also has a nice intuitive interpretation: we can view the $i$-th row of $\boldsymbol{V}$ as a length-$K$ feature vector describing item $i$.

Unfortunately, the symmetry requirement limits the types of correlations that a DPP can capture. A symmetric model is able to capture only nonpositive interactions between items, since $\boldsymbol{L}_{ij}\boldsymbol{L}_{ji} = \boldsymbol{L}_{ij}^2 \geq 0$, whereas a nonsymmetric $\boldsymbol{L}$ can also capture positive correlations. (Again, see Gartrell et al. (2019, Section 2.1) for more intuition.) To expand coverage to nonsymmetric matrices in $P_0$, it is natural to consider nonsymmetric PSD matrices. In what follows, we denote by $P_0^+$ the set of all nonsymmetric (and symmetric) PSD matrices. Any nonsymmetric PSD matrix is in $P_0$ (Gartrell et al., 2019, Lemma 1), so $P_0^+ \subseteq P_0$. However, unlike in the symmetric case, the set of nonsymmetric PSD

---

[*]Recall that a matrix $\boldsymbol{L} \in \mathbb{R}^{M \times M}$ is defined to be PSD if and only if $\boldsymbol{x}^\top \boldsymbol{L} \boldsymbol{x} \geq 0$, for all $\boldsymbol{x} \in \mathbb{R}^M$.

matrices does not fully cover the set of nonsymmetric $P_0$ matrices. For example, consider

$$L = \begin{pmatrix} 1 & 5/3 \\ 1/2 & 1 \end{pmatrix} \text{ with } \det(L_{\{1\}}), \det(L_{\{2\}}), \det(L_{\{1,2\}}) \geq 0, \text{ but } x^\top L x < 0 \text{ for } x = \begin{pmatrix} -1 \\ 1 \end{pmatrix}.$$

Still, nonsymmetric PSD matrices cover a large enough portion of the $P_0$ space to be useful in practice, as evidenced by the experiments of Gartrell et al. (2019). This work covered the $P_0^+$ space by using the following decomposition: $L := S + A$, with $S := V V^\top$ for $V \in \mathbb{R}^{M \times K}$, and $A := BC^\top - CB^\top$ for $B, C \in \mathbb{R}^{M \times K}$. This decomposition makes use of the fact that any matrix $L$ can be decomposed uniquely as the sum of a symmetric matrix $S = (L + L^T)/2$ and a skew-symmetric matrix $A = (L - L^T)/2$. All skew-symmetric matrices $A$ are trivially PSD, since $x^\top A x = 0$ for all $x \in \mathbb{R}^M$. Hence, the $L$ here is guaranteed to be PSD simply because its $S$ uses the standard Gramian decomposition $V V^\top$.

In this work we will also only consider $P_0^+$, and leave to future work the problem of finding tractable ways to cover the rest of $P_0$. We propose a new decomposition of $L$ that also covers the $P_0^+$ space, but allows for more scalable learning. As in prior work, our decomposition has inner dimension $K$ that could be as large as $M$, but is usually much smaller in practice. Our algorithms work well for modest values of $K$. In cases where the natural $K$ is larger (e.g., natural language processing), random projections can often be used to significantly reduce $K$ (Gillenwater et al., 2012a).

## 3 NEW KERNEL DECOMPOSITION AND SCALABLE LEARNING

Prior work on NDPPs proposed a maximum likelihood estimation (MLE) algorithm (Gartrell et al., 2019). Due to that work's particular kernel decomposition, this algorithm had complexity cubic in the number of items $M$. Here, we propose a kernel decomposition that reduces this to linear in $M$.

We begin by showing that our new decomposition covers the space of $P_0^+$ matrices. Before diving in, let us define $\Sigma_i := \begin{pmatrix} 0 & \lambda_i \\ -\lambda_i & 0 \end{pmatrix}$ as shorthand for a $2 \times 2$ block matrix with zeros on-diagonal and opposite values off-diagonal. Then, our proposed decomposition is as follows:

$$L := S + A, \text{ with } S := V V^\top \text{ and } A := BCB^\top, \tag{1}$$

where $V, B \in \mathbb{R}^{M \times K}$, and $C \in \mathbb{R}^{K \times K}$ is a block-diagonal matrix with some diagonal blocks of the form $\Sigma_i$, with $\lambda_i > 0$, and zeros elsewhere. The following lemma shows that this decomposition covers the space of $P_0^+$ matrices.

**Lemma 1.** *Let $A \in \mathbb{R}^{M \times M}$ be a skew-symmetric matrix with rank $\ell \leq M$. Then, there exist $B \in \mathbb{R}^{M \times \ell}$ and positive numbers $\lambda_1, \ldots, \lambda_{\lfloor \ell/2 \rfloor}$, such that $A = BCB^\top$, where $C \in \mathbb{R}^{\ell \times \ell}$ is the block-diagonal matrix with $\lfloor \ell/2 \rfloor$ diagonal blocks of size 2 given by $\Sigma_i$, $i = 1, \ldots, \lfloor \ell/2 \rfloor$ and zero elsewhere.*

The proof of Lemma 1 and all subsequent results can be found in Appendix F. With this decomposition in hand, we now proceed to show that it can be used for linear-time MLE learning. To do so, we must show that corresponding NDPP log-likelihood objective and gradient can be computed in time linear in $M$. Given a collection of $n$ observed subsets $\{Y_1, ..., Y_n\}$ composed of items from $\mathcal{Y} = \llbracket M \rrbracket$, the full formulation of the regularized log-likelihood is:

$$\phi(V, B, C) = \frac{1}{n} \sum_{i=1}^n \log \det \left( V_{Y_i} V_{Y_i}^\top + B_{Y_i} C B_{Y_i}^\top \right) - \log \det \left( V V^\top + BCB^\top + I \right) - R(V, B), \tag{2}$$

where $V_{Y_i} \in \mathbb{R}^{|Y_i| \times K}$ denotes a matrix composed of the rows of $V$ that correspond to the items in $Y_i$. The regularization term, $R(V, B)$, is defined as follows:

$$R(V, B) = \alpha \sum_{i=1}^M \frac{1}{\mu_i} \|v_i\|_2^2 + \beta \sum_{i=1}^M \frac{1}{\mu_i} \|b_i\|_2^2, \tag{3}$$

where $\mu_i$ counts the number of occurrences of item $i$ in the training set, $v_i$ and $b_i$ are rows of $V$ and $B$, respectively, and $\alpha, \beta > 0$ are tunable hyperparameters. This regularization is similar to that of prior works (Gartrell et al., 2017; 2019). We omit regularization for $C$.

Theorem 1 shows that computing the regularized log-likelihood and its gradient both have time complexity linear in $M$. The complexities also depend on $K$, the rank of the NDPP, and $K'$, the size of the largest observed subset in the data. For many real-world datasets we observe that $K' \ll M$, and we set $K = K'$. Hence, linearity in $M$ means that we can efficiently perform learning for datasets with very large ground sets, which is impossible with the cubic-complexity $\boldsymbol{L}$ decomposition in prior work (Gartrell et al., 2019).

**Theorem 1.** *Given an NDPP with kernel $\boldsymbol{L} = \boldsymbol{V}\boldsymbol{V}^\top + \boldsymbol{B}\boldsymbol{C}\boldsymbol{B}^\top$, parameterized by $\boldsymbol{V}$ of rank $K$, $\boldsymbol{B}$ of rank $K$, and a $K \times K$ matrix $\boldsymbol{C}$, we can compute the regularized log-likelihood (Eq. 2) and its gradient in $O(MK^2 + K^3 + nK'^3)$ time, where $K'$ is the size of the largest of the $n$ training subsets.*

## 4 MAP INFERENCE

After learning an NDPP, one can then use it to infer the most probable item subsets in various situations. Several inference algorithms have been well-studied for symmetric DPPs, including sampling (Kulesza & Taskar, 2011; Anari et al., 2016; Li et al., 2016; Launay et al., 2018; Gillenwater et al., 2019; Poulson, 2019; Dereziński, 2019) and MAP inference (Gillenwater et al., 2012b; Han et al., 2017; Chen et al., 2018; Han & Gillenwater, 2020). We focus on MAP inference:

$$\underset{Y \subseteq \mathcal{Y}}{\operatorname{argmax}} \det(\boldsymbol{L}_Y) \quad \text{such that} \quad |Y| = k, \tag{4}$$

for cardinality budget $k \le M$. MAP inference is a better fit than sampling when the end application requires the generation of a single output set, which is usually the case in practice (e.g., this is usually true for recommender systems). MAP inference for DPPs is known to be NP-hard even in the symmetric case (Ko et al., 1995; Kulesza et al., 2012). For symmetric DPPs, one usually approximates the MAP via the standard greedy algorithm for submodular maximization (Nemhauser et al., 1978). First, we describe how to efficiently implement this for NDPPs. Then, in Section 4.1 we prove a lower bound on its approximation quality. To the best of our knowledge, this is the first investigation of how to apply the greedy algorithm to NDPPs.

Greedy begins with an empty set and repeatedly adds the item that maximizes the marginal gain until the chosen set is size $k$. Here, we design an efficient greedy algorithm for the case where the NDPP kernel is low-rank. For generality, in what follows we write the kernel as $\boldsymbol{L} = \boldsymbol{B}\boldsymbol{C}\boldsymbol{B}^\top$, since one can easily rewrite our matrix decomposition (Eq. 1), as well as that of Gartrell et al. (2019), to take this form. For example, for our decomposition: $\boldsymbol{L} = \boldsymbol{V}\boldsymbol{V}^\top + \boldsymbol{B}\boldsymbol{C}\boldsymbol{B}^\top = (\boldsymbol{V} \quad \boldsymbol{B}) \begin{pmatrix} \boldsymbol{I} & \boldsymbol{0} \\ \boldsymbol{0} & \boldsymbol{C} \end{pmatrix} \begin{pmatrix} \boldsymbol{V}^\top \\ \boldsymbol{B}^\top \end{pmatrix}$.

Using Schur's determinant identity, we first observe that, for $Y \subseteq [\![M]\!]$ and $i \in [\![M]\!]$, the marginal gain of a NDPP can be written as

$$\frac{\det(\boldsymbol{L}_{Y \cup \{i\}})}{\det(\boldsymbol{L}_Y)} = \boldsymbol{L}_{ii} - \boldsymbol{L}_{iY}(\boldsymbol{L}_Y)^{-1}\boldsymbol{L}_{Yi} = \boldsymbol{b}_i\boldsymbol{C}\boldsymbol{b}_i^\top - \boldsymbol{b}_i\boldsymbol{C} \left( \boldsymbol{B}_Y^\top (\boldsymbol{B}_Y\boldsymbol{C}\boldsymbol{B}_Y^\top)^{-1}\boldsymbol{B}_Y \right) \boldsymbol{C}\boldsymbol{b}_i^\top, \tag{5}$$

where $\boldsymbol{b}_i \in \mathbb{R}^{1 \times K}$ and $\boldsymbol{B}_Y \in \mathbb{R}^{|Y| \times K}$. A naïve computation of Eq. 5 is $O(K^2 + k^3)$, since we must invert a $|Y| \times |Y|$ matrix, where $|Y| \le k$. However, one can compute Eq. 5 more efficiently by observing that its $\boldsymbol{B}_Y^\top (\boldsymbol{B}_Y\boldsymbol{C}\boldsymbol{B}_Y^\top)^{-1}\boldsymbol{B}_Y$ component can actually be expressed without an inverse, as a rank-$|Y|$ matrix, that can be computed in $O(K^2)$ time.

**Lemma 2.** *Given $\boldsymbol{B} \in \mathbb{R}^{M \times K}$, $\boldsymbol{C} \in \mathbb{R}^{K \times K}$, and $Y = \{a_1, \ldots, a_k\} \subseteq [\![M]\!]$, let $\boldsymbol{b}_i \in \mathbb{R}^{1 \times K}$ be the $i$-th row in $\boldsymbol{B}$ and $\boldsymbol{B}_Y \in \mathbb{R}^{|Y| \times K}$ be a matrix containing rows in $\boldsymbol{B}$ indexed by $Y$. Then, it holds that*

$$\boldsymbol{B}_Y^\top (\boldsymbol{B}_Y\boldsymbol{C}\boldsymbol{B}_Y^\top)^{-1}\boldsymbol{B}_Y = \sum_{j=1}^{k} \boldsymbol{p}_j^\top \boldsymbol{q}_j, \tag{6}$$

*where row vectors $\boldsymbol{p}_j, \boldsymbol{q}_j \in \mathbb{R}^{1 \times K}$ for $j = 1, \ldots, k$ satisfy $\boldsymbol{p}_1 = \boldsymbol{b}_{a_1}/(\boldsymbol{b}_{a_1}\boldsymbol{C}\boldsymbol{b}_{a_1}^\top)$, $\boldsymbol{q}_1 = \boldsymbol{b}_{a_1}$, and*

$$\boldsymbol{p}_{j+1} = \frac{\boldsymbol{b}_{a_j} - \boldsymbol{b}_{a_j}\boldsymbol{C}^\top \sum_{i=1}^{j} \boldsymbol{q}_i^\top \boldsymbol{p}_i}{\boldsymbol{b}_{a_j}\boldsymbol{C}(\boldsymbol{b}_{a_j} - \boldsymbol{b}_{a_j}\boldsymbol{C}^\top \sum_{i=1}^{j} \boldsymbol{q}_i^\top \boldsymbol{p}_i)^\top}, \qquad \boldsymbol{q}_{j+1} = \boldsymbol{b}_{a_j} - \boldsymbol{b}_{a_j}\boldsymbol{C} \sum_{i=1}^{j} \boldsymbol{p}_i^\top \boldsymbol{q}_i. \tag{7}$$

---

**Algorithm 1** Greedy MAP inference/conditioning for low-rank NDPPs

---

1: **Input:** $\boldsymbol{B} \in \mathbb{R}^{M \times K}$, $\boldsymbol{C} \in \mathbb{R}^{K \times K}$, the cardinality $k$ $\qquad \triangleright$ And $\{a_1, \ldots, a_k\}$ for conditioning
2: **initialize** $\boldsymbol{P} \leftarrow [\,]$, $\boldsymbol{Q} \leftarrow [\,]$ and $Y \leftarrow \emptyset$
3: $\Delta_i \leftarrow \boldsymbol{b}_i \boldsymbol{C} \boldsymbol{b}_i^\top$ for $i \in [\![M]\!]$ where $\boldsymbol{b}_i \in \mathbb{R}^{1 \times K}$ is the $i$-th row in $\boldsymbol{B}$
4: $a \leftarrow \operatorname{argmax}_i \Delta_i$ and $Y \leftarrow Y \cup \{a\}$ $\qquad\qquad \triangleright a \leftarrow a_1$ for conditioning
5: **while** $|Y| \leq k$ **do**
6: $\qquad \boldsymbol{p} \leftarrow \left(\boldsymbol{b}_a - \boldsymbol{b}_a \boldsymbol{C}^\top \boldsymbol{Q}^\top \boldsymbol{P}\right) / \Delta_a$
7: $\qquad \boldsymbol{q} \leftarrow \boldsymbol{b}_a - \boldsymbol{b}_a \boldsymbol{C} \boldsymbol{P}^\top \boldsymbol{Q}$
8: $\qquad \boldsymbol{P} \leftarrow [\boldsymbol{P}; \boldsymbol{p}]$ and $\boldsymbol{Q} \leftarrow [\boldsymbol{Q}; \boldsymbol{q}]$
9: $\qquad \Delta_i \leftarrow \Delta_i - \left(\boldsymbol{b}_i \boldsymbol{C} \boldsymbol{p}^\top\right)\left(\boldsymbol{b}_i \boldsymbol{C}^\top \boldsymbol{q}^\top\right)$ for $i \in [\![M]\!], i \notin Y$
10: $\qquad a \leftarrow \operatorname{argmax}_i \Delta_i$ and $Y \leftarrow Y \cup \{a\}$ $\qquad\quad \triangleright a \leftarrow a_{|Y|+1}$ for conditioning
11: **end while**
12: **return** $Y$ $\qquad\qquad\qquad\qquad\qquad\qquad \triangleright$ **return** $\{\Delta_i\}_{i=1}^M$ for conditioning

---

Table 1: Algorithm complexities for several DPP models. Our model and the symmetric DPP model (Gartrell et al., 2017) can perform both tasks in time linear in the size of ground set $M$, but ours is a more general model that can capture positive as well as negative item correlations.

| Low-rank DPP Models | MLE Learning Runtime | MAP Inference Runtime | MLE Learning Memory | MAP Inference Memory |
|---|---|---|---|---|
| Symmetric DPP (Gartrell et al., 2017) | $O(MK^2 + nK^3)$ | $O(MKk + MK^2)$ | $O(MK)$ | $O(MK)$ |
| Nonsymmetric DPP (Gartrell et al., 2019) | $O(M^3 + MK^2 + nK^3)$ | $O(MKk + MK^2)$ | $O(M^2)$ | $O(MK)^\dagger$ |
| Scalable nonsymmetric DPP (this work) | $O(MK^2 + nK^3)$ | $O(MKk + MK^2)$ | $O(MK + K^2)$ | $O(MK + K^2)$ |

Plugging Eq. 6 into Eq. 5, the marginal gain with respect to $Y \cup \{a\}$ can be computed by simply updating from the previous gain with respect to $Y$. That is,

$$\frac{\det(\boldsymbol{L}_{Y \cup \{a,i\}})}{\det(\boldsymbol{L}_{Y \cup \{a\}})} = \boldsymbol{b}_i \boldsymbol{C} \boldsymbol{b}_i^\top - \sum_{j=1}^{|Y|+1} \left(\boldsymbol{b}_i \boldsymbol{C} \boldsymbol{p}_j^\top\right)\left(\boldsymbol{b}_i \boldsymbol{C}^\top \boldsymbol{q}_j^\top\right) \tag{8}$$

$$= \frac{\det(\boldsymbol{L}_{Y \cup \{i\}})}{\det(\boldsymbol{L}_Y)} - \left(\boldsymbol{b}_i \boldsymbol{C} \boldsymbol{p}_{|Y|+1}^\top\right)\left(\boldsymbol{b}_i \boldsymbol{C}^\top \boldsymbol{q}_{|Y|+1}^\top\right). \tag{9}$$

The marginal gains when $Y = \emptyset$ are equal to diagonals of $\boldsymbol{L}$ and require $O(MK^2)$ operations. Then, computing the update terms in Eq. 9 for all $i \in [\![M]\!]$ needs $O(MK)$ operations. Since the total number of updates is $k$, the overall complexity becomes $O(MK^2 + MKk)$. We provide a full description of the implied greedy algorithm for low-rank NDPPs in Algorithm 1.

Table 1 summarizes the complexitiy of our methods and those of previous work. Note that the full $M \times M$ $\boldsymbol{L} + \boldsymbol{I}$ matrix is used to compute the DPP normalization constant in Gartrell et al. (2019), which is why this approach has memory complexity of $O(M^2)$ for MLE learning.

### 4.1 Approximation Guarantee for Greedy NDPP MAP Inference

As mentioned above, Algorithm 1 is an instantiation of the standard greedy algorithm used for submodular maximization (Nemhauser et al., 1978). This algorithm has a $(1 - 1/e)$-approximation guarantee for the problem of maximizing nonnegative, monotone submodular functions. While the function $f(Y) = \log \det(\boldsymbol{L}_Y)$ is submodular for a symmetric PSD $\boldsymbol{L}$ (Kelmans & Kimelfeld, 1983), it is not monotone. Often, as in Han & Gillenwater (2020), it is assumed that the smallest eigenvalue of $\boldsymbol{L}$ is greater than 1, which guarantees montonicity. There is no particular evidence that this assumption is true for practical models, but nevertheless the greedy algorithm tends to

---

$\dagger$ The exact memory complexity for MAP inference is $3MK$, since $\boldsymbol{V}$, $\boldsymbol{B}$, and $\boldsymbol{C}$ used in this model are all $M \times K$ matrices.

perform well in practice for symmetric DPPs. Here, we prove a similar approximation guarantee that covers NDPPs as well, even though the function $f(Y) = \log \det(\boldsymbol{L}_Y)$ is non-submodular when $\boldsymbol{L}$ is nonsymmetric. In Section 5.5, we further observe that, as for symmetric DPPs, the greedy algorithm seems to work well in practice for NDPPs.

We leverage a recent result of Bian et al. (2017), who proposed an extension of greedy algorithm guarantees to non-submodular functions. Their result is based on the submodularity ratio and curvature of the objective function, which measure to what extent it has submodular properties. Theorem 2 extends this to provide an approximation ratio for greedy MAP inference of NDPPs.

**Theorem 2.** *Consider a nonsymmetric low-rank DPP $\boldsymbol{L} = \boldsymbol{V}\boldsymbol{V}^\top + \boldsymbol{B}\boldsymbol{C}\boldsymbol{B}^\top$, where $\boldsymbol{V}, \boldsymbol{B}$ are of rank $K$, and $\boldsymbol{C} \in \mathbb{R}^{K \times K}$. Given a cardinality budget $k$, let $\sigma_{\min}$ and $\sigma_{\max}$ denote the smallest and largest singular values of $\boldsymbol{L}_Y$ for all $Y \subseteq \llbracket M \rrbracket$ and $|Y| \leq 2k$. Assume that $\sigma_{\min} > 1$. Then,*

$$\log \det(\boldsymbol{L}_{Y^G}) \geq \frac{4(1 - e^{-1/4})}{2(\log \sigma_{\max}/\log \sigma_{\min}) - 1} \log \det(\boldsymbol{L}_{Y^*}) \tag{10}$$

*where $Y^G$ is the output of Algorithm 1 and $Y^*$ is the optimal solution of MAP inference in Eq. 4.*

Thus, when the kernel has a small value of $\log \sigma_{\max}/\log \sigma_{\min}$, the greedy algorithm finds a near-optimal solution. In practice, we observe that the greedy algorithm finds a near-optimal solution even for large values of this ratio (see Section 5.5). As remarked above, there is no evidence that the condition $\sigma_{\min} > 1$ is usually true in practice. While this condition can be achieved by multiplying $\boldsymbol{L}$ by a constant, this leads to a (potentially large) additive term in Eq. 10. We provide Corollary 1 in Appendix D, which excludes the $\sigma_{\min} > 1$ assumption, and quantifies this additive term.

### 4.2 GREEDY CONDITIONING FOR NEXT-ITEM PREDICTION

We briefly describe here a small modification to the greedy algorithm that is necessary if one wants to use it as a tool for next-item prediction. Given a set $Y \subseteq \llbracket M \rrbracket$, Kulesza et al. (2012) showed that a DPP with $\boldsymbol{L}$ conditioned on the inclusion of the items in $Y$ forms another DPP with kernel $\boldsymbol{L}^Y := \boldsymbol{L}_{\bar{Y}} - \boldsymbol{L}_{\bar{Y},Y}\boldsymbol{L}_Y^{-1}\boldsymbol{L}_{\bar{Y},Y}$ where $\bar{Y} = \llbracket M \rrbracket \setminus Y$. The singleton probability $\Pr(Y \cup \{i\} \mid Y) \propto \boldsymbol{L}_{ii}^Y$ can be useful for doing next-item prediction. We can use the same machinery from the greedy algorithm's marginal gain computations to effectively compute these singletons. More concretely, suppose that we are doing next-item prediction as a shopper adds items to a digital cart. We predict the item that maximizes the marginal gain, conditioned on the current cart contents (the set $Y$). When the shopper adds the next item to the cart, we update $Y$ to include this item, rather than our predicted item (line 10 in Algorithm 1). We then iterate until the shopper checks out. The comments on the righthand side of Algorithm 1 summarize this procedure. The runtime of this prediction is the same that of the greedy algorithm, $O(MK^2 + MK|Y|)$. We note that this cost is comparable to that of an approach based on the DPP dual kernel from prior work (Mariet et al., 2019), which has $O(MK^2 + K^3 + |Y|^3)$ complexity. However, since it is non-trivial to define the dual kernel for NDPPs, the greedy algorithm may be the simpler choice for next-item prediction for NDPPs.

## 5 EXPERIMENTS

To further simplify learning and MAP inference, we set $\boldsymbol{B} = \boldsymbol{V}$, which results in $\boldsymbol{L} = \boldsymbol{V}\boldsymbol{V}^\top + \boldsymbol{V}\boldsymbol{C}\boldsymbol{V}^\top = \boldsymbol{V}(\boldsymbol{I} + \boldsymbol{C})\boldsymbol{V}^\top$. This change also simplifies regularization, so that we only perform regularization on $\boldsymbol{V}$, as indicated in the first term of Eq. 3, leaving us with the single regularization hyperparameter of $\alpha$. While setting $\boldsymbol{B} = \boldsymbol{V}$ restricts the class of nonsymmetric $\boldsymbol{L}$ kernels that can be represented, we compensate for this restriction by relaxing the block-diagonal structure imposed on $\boldsymbol{C}$, so that we learn a full skew-symmetric $K \times K$ matrix $\boldsymbol{C}$. To ensure that $\boldsymbol{C}$ and thus $\boldsymbol{A}$ is skew-symmetric, we parameterize $\boldsymbol{C}$ by setting $\boldsymbol{C} = \boldsymbol{D} - \boldsymbol{D}^T$, were $\boldsymbol{D}$ varies over $\mathbb{R}^{K \times K}$.

Code for all experiments is available at
https://github.com/cgartrel/scalable-nonsymmetric-DPPs.

### 5.1 DATASETS

We perform experiments on several real-world public datasets composed of subsets:

1. **Amazon Baby Registries:** This dataset consists of registries or "baskets" of baby products, and has been used in prior work on DPP learning (Gartrell et al., 2016; 2019; Gillenwater et al., 2014; Mariet & Sra, 2015). The registries contain items from 15 different categories, such as "apparel", with a catalog of up to 100 items per category. Our evaluation mirrors that of Gartrell et al. (2019); we evaluate on the popular apparel category, which contains 14,970 registries, as well as on a dataset composed of the three most popular categories: apparel, diaper, and feeding, which contains a total of 31,218 registries.

2. **UK Retail:** This dataset (Chen et al., 2012) contains baskets representing transactions from an online retail company that sells unique all-occasion gifts. We omit baskets with more than 100 items, leaving us with a dataset containing 19,762 baskets drawn from a catalog of $M = 3{,}941$ products. Baskets containing more than 100 items are in the long tail of the basket-size distribution of the data, so omitting larger baskets is reasonable, and allows us to use a low-rank factorization of the DPP with $K = 100$.

3. **Instacart:** This dataset (Instacart, 2017) contains baskets purchased by Instacart users. We omit baskets with more than 100 items, resulting in 3.2 million baskets and a catalog of 49,677 products.

4. **Million Song:** This dataset (McFee et al., 2012) contains playlists ("baskets") of songs played by Echo Nest users. We trim playlists with more than 150 items, leaving us with 968,674 baskets and a catalog of 371,410 songs.

## 5.2 EXPERIMENTAL SETUP AND METRICS

We use a small held-out validation set, consisting of 300 randomly-selected baskets, for tracking convergence during training and for tuning hyperparameters. A random selection of 2000 of the remaining baskets are used for testing, and the rest are used for training. Convergence is reached during training when the relative change in validation log-likelihood is below a predetermined threshold. We use PyTorch with Adam (Kingma & Ba, 2015) for optimization. We initialize $C$ from the standard Gaussian distribution with mean 0 and variance 1, and $B$ (which we set equal to $V$) is initialized from the uniform$(0, 1)$ distribution.

**Subset expansion task.** We use greedy conditioning to do next-item prediction, as described in Section 4.2. We compare methods using a standard recommender system metric: mean percentile rank (MPR) (Hu et al., 2008; Li et al., 2010). MPR of 50 is equivalent to random selection; MPR of 100 means that the model perfectly predicts the next item. See Appendix A for a complete description of the MPR metric.

**Subset discrimination task.** We also test the ability of a model to discriminate observed subsets from randomly generated ones. For each subset in the test set, we generate a subset of the same length by drawing items uniformly at random (and we ensure that the same item is not drawn more than once for a subset). We compute the AUC for the model on these observed and random subsets, where the score for each subset is the log-likelihood that the model assigns to the subset.

## 5.3 PREDICTIVE PERFORMANCE RESULTS FOR LEARNING

Since the focus of our work is on improving NDPP scalability, we use the low-rank symmetric DPP (Gartrell et al., 2017) and the low-rank NDPP of prior work (Gartrell et al., 2019) as baselines for our experiments. Table 2 compares these approaches and our scalable low-rank NDPP. We see that NDPPs generally outperform symmetric DPPs. Furthermore, we see that our scalable NDPP matches or exceeds the predictive quality of the baseline NDPP. We believe that our model sometimes improves upon this baseline NDPP due to the use of a simpler kernel decomposition with fewer parameters, likely leading to a simplified optimization landscape.

## 5.4 TIME COMPARISON FOR LEARNING

In Fig. 1, we report the wall-clock training time of the decomposition of Gartrell et al. (2019) (NDPP) and our scalable NDPP for the Amazon: 3-category (Fig. 1(a)) and UK Retail (Fig. 1(b)) datasets. As expected, we observe that the scalable NDPP trains far faster than the NDPP for datasets with large ground sets. For the Amazon: 3-category dataset, both approaches show comparable results, with the scalable NDPP converging $1.07\times$ faster than NDPP. But for the UK Retail dataset, which has a much larger ground set, our scalable NDPP achieves convergence about $8.31\times$ faster. Notice

Table 2: Average MPR, AUC, and test log-likelihood for all datasets, for the low-rank symmetric DPP (Gartrell et al., 2017), low-rank NDPP (Gartrell et al., 2019), and our scalable NDPP models. MPR and AUC results show 95% confidence estimates obtained via bootstrapping. Bold values indicate improvement over the symmetric low-rank DPP outside of the confidence interval. See Appendix B for the hyperparameter settings used in these experiments. The baseline NDPP model cannot be feasibly trained on the Instacart and Million Song datasets, as memory and computational costs are prohibitive due to large ground set sizes.

| | **Amazon: Apparel** ($M = 100$) | | | **Amazon: 3-category** ($M = 300$) | | |
|---|---|---|---|---|---|---|
| Metric | Sym | Nonsym | Scalable nonsym | Sym | Nonsym | Scalable nonsym |
| MPR | $62.63 \pm 1.81$ | $\mathbf{72.20 \pm 3.07}$ | $69.02 \pm 2.57$ | $61.0 \pm 2.73$ | $\mathbf{74.10 \pm 2.49}$ | $\mathbf{73.04 \pm 2.58}$ |
| AUC | $0.68 \pm 0.05$ | $\mathbf{0.77 \pm 0.03}$ | $0.74 \pm 0.03$ | $0.76 \pm 0.03$ | $\mathbf{0.82 \pm 0.02}$ | $\mathbf{0.82 \pm 0.02}$ |
| test log-likelihood | -10.02 | **-9.64** | **-9.63** | -18.11 | **-16.96** | **-17.14** |

| | **UK Retail** ($M = 3{,}941$) | | | **Instacart** ($M = 49{,}677$) | | |
|---|---|---|---|---|---|---|
| Metric | Sym | Nonsym | Scalable nonsym | Sym | Nonsym | Scalable nonsym |
| MPR | $69.95 \pm 1.32$ | $\mathbf{74.17 \pm 1.37}$ | $\mathbf{76.79 \pm 1.17}$ | $93.86 \pm 0.55$ | - | $93.13 \pm 0.53$ |
| AUC | $0.58 \pm 0.01$ | $\mathbf{0.66 \pm 0.01}$ | $\mathbf{0.73 \pm 0.01}$ | $0.83 \pm 0.01$ | - | $\mathbf{0.85 \pm 0.005}$ |
| test log-likelihood | -116.23 | **-104.38** | **-100.65** | -72.81 | - | **-72.74** |

| | **Million Song** ($M = 371{,}410$) | | |
|---|---|---|---|
| Metric | Sym | Nonsym | Scalable nonsym |
| MPR | $90.37 \pm 0.71$ | - | $90.41 \pm 0.75$ |
| AUC | $0.69 \pm 0.01$ | - | $\mathbf{0.77 \pm 0.01}$ |
| test log-likelihood | -335.25 | - | **-317.16** |

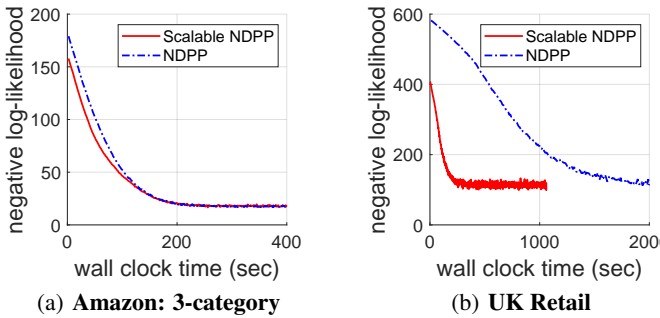

(a) **Amazon: 3-category**    (b) **UK Retail**

Figure 1: The negative log-likelihood of the training set for Gartrell et al. (2019)'s NDPP (blue, dashed) and our scalable NDPP (red, solid) versus wall-clock time for the (a) Amazon: 3-category and (b) UK Retail datasets.

that our scalable NDPP also opens to the door to training on datasets with large $M$, such as the Instacart and Million Song dataset, which is infeasible for the baseline NDPP due to high memory and compute costs. For example, NDPP learning using Gartrell et al. (2019) for the Million Song dataset would require approximately 1.1 TB of memory, while using our scalable NDPP approach requires approximately 445.9 MB.

## 5.5 PERFORMANCE RESULTS FOR MAP INFERENCE

We run various approximatation algorithms for MAP inference, including the greedy algorithm (Algorithm 1), stochastic greedy algorithm (Mirzasoleiman et al., 2015), MCMC-based DPP sampling (Li et al., 2016), and greedy local search (Kathuria & Deshpande, 2016). The stochastic greedy algorithm computes marginal gains of a few items chosen uniformly at random and selects the best among them. The MCMC sampling begins with a random subset $Y$ of size $k$ and picks $i \in Y$ and $j \notin Y$ uniformly at random. Then, it swaps them with probability $\det(\boldsymbol{L}_{Y \cup \{j\} \setminus \{i\}})/(\det(\boldsymbol{L}_{Y \cup \{j\} \setminus \{i\}}) + \det(\boldsymbol{L}_Y))$ and iterates this process. The greedy local search algorithm (Kathuria & Deshpande, 2016) starts from the output from the greedy algorithm, $Y^G$, and replaces $i \in Y^G$ with $j \notin Y^G$ that gives the maximum improvement, if such $i, j$ exist. This replacement process iterates until no improvement exists, or at

Table 3: Average relative error and 95% confidence intervals of MAP inference algorithms on NDPPs learned from real-world datasets. For all datasets, we evaluate 10 kernels learned with different initializations, and run 100 random trials for stochastic greedy (Mirzasoleiman et al., 2015) and MCMC sampling (Li et al., 2016). All errors are relative to greedy local search (Kathuria & Deshpande, 2016).

| Algorithms | Amazon: Apparel | Amazon: 3-category | UK Retail | Instacart | Million Song |
|---|---|---|---|---|---|
| Greedy (Algorithm 1) | **0.0336 ± 0.0066** | **0.0093 ± 0.0015** | **0.0446 ± 0.0035** | **0.0173 ± 0.0028** | **0.0052 ± 0.0017** |
| Stochastic greedy | 0.1606 ± 0.0133 | 0.1838 ± 0.0116 | 0.0960 ± 0.0078 | 0.1229 ± 0.0091 | 0.0823 ± 0.0108 |
| MCMC sampling | 0.7155 ± 0.0287 | 0.7094 ± 0.0207 | 0.9365 ± 0.0342 | 1.9291 ± 0.047 | 1.0493 ± 0.0607 |

Table 4: Wall-clock time (in milliseconds) of MAP inference algorithms on NDPPs learned from real-world datasets.

| Algorithms | Amazon: Apparel | Amazon: 3-category | UK Retail | Instacart | Million Song |
|---|---|---|---|---|---|
| Greedy local search | 5.78 ms | 9.67 ms | 58.74 ms | 1.024 s | 7.277 s |
| Greedy (Algorithm 1) | **0.14 ms** | **0.34 ms** | **1.60 ms** | **36.16 ms** | 338.09 ms |
| Stochastic greedy | 0.25 ms | 0.47 ms | 1.79 ms | 36.94 ms | 348.67 ms |
| MCMC sampling | 0.19 ms | 0.35 ms | 2.85 ms | 42.85 ms | **303.20 ms** |

most $k^2 \log(10k)$ steps have been completed, to guarantee a tight approximation (Kathuria & Deshpande, 2016). We use greedy local search as a baseline since it always returns a better solution than greedy. However, it is the slowest among all algorithms, as its time complexity is $O(MKk^4 \log k)$. We choose $k = 10$, and provide more details of all algorithms in Appendix C.

To evaluate the performance of MAP inference, we report the relative log-determinant ratio defined as

$$\left| \frac{\log \det(\boldsymbol{L}_{Y^*}) - \log \det(\boldsymbol{L}_Y)}{\log \det(\boldsymbol{L}_{Y^*})} \right|$$

where $Y$ is the output of benchmark algorithms and $Y^*$ is the greedy local search result. Results are reported in Table 3. We observe that the greedy (Algorithm 1) achieves performance close to that of the significantly more expensive greedy local search algorithm, with relative errors of up to $0.045$. Stochastic greedy and MCMC sampling have significantly larger errors.

For completeness, in Appendix E we also present experiments comparing the performance of greedy and exact MAP on small synthetic NDPPs, for which the exact MAP can be feasibly computed.

## 5.6 TIME COMPARISON FOR MAP INFERENCE

We provide the wall-clock time of the above algorithms for real-world datasets in Table 4. Observe that the greedy algorithm is the fastest method for all datasets except Million Song. For Million Song, MCMC sampling is faster than other approaches, but it has much larger relative errors in terms of log-determinant (see Table 3), which is not suitable for our purposes.

## 6 CONCLUSION

We have presented a new decomposition for nonsymmetric DPP kernels that can be learned in time linear in the size of the ground set, which is a significant improvement over the complexity of prior work. Empirical results indicate that this decomposition matches the predictive performance of the prior decomposition. We have also derived the first MAP algorithm for nonsymmetric DPPs and proved a lower bound on the quality of its approximation. In future work we hope to develop intuition about the meaning of the parameters in the $\boldsymbol{C}$ matrix and consider kernel decompositions that cover other parts of the nonsymmetric $P_0$ space.

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

## A  MEAN PERCENTILE RANK

We begin our definition of MPR by defining percentile rank (PR). First, given a set $J$, let $p_{i,J} = \Pr(J \cup \{i\} \mid J)$. The percentile rank of an item $i$ given a set $J$ is defined as

$$\text{PR}_{i,J} = \frac{\sum_{i' \notin J} \mathbb{1}(p_{i,J} \ge p_{i',J})}{|\mathcal{Y} \backslash J|} \times 100\%$$

where $\mathcal{Y} \backslash J$ indicates those elements in the ground set $\mathcal{Y}$ that are not found in $J$.

For our evaluation, given a test set $Y$, we select a random element $i \in Y$ and compute $\text{PR}_{i,Y \backslash \{i\}}$. We then average over the set of all test instances $\mathcal{T}$ to compute the mean percentile rank (MPR):

$$\text{MPR} = \frac{1}{|\mathcal{T}|} \sum_{Y \in \mathcal{T}} \text{PR}_{i,Y \backslash \{i\}}.$$

## B  HYPERPARAMETERS FOR EXPERIMENTS IN TABLE 2

**Preventing numerical instabilities**: The first term on the right side of Eq. 2 will be singular whenever $|Y_i| > K$, where $Y_i$ is an observed subset. Therefore, to address this in practice we set $K$ to the size of the largest subset observed in the data, $K'$, as in Gartrell et al. (2017). However, this does not entirely address the issue, as the first term on the right side of Eq. 2 may still be singular even when $|Y_i| \le K$. In this case though, we know that we are not at a maximum, since the value of the objective function is $-\infty$. Numerically, to prevent such singularities, in our implementation we add a small $\epsilon \boldsymbol{I}$ correction to each $\boldsymbol{L}_{Y_i}$ when optimizing Eq. 2 (we set $\epsilon = 10^{-5}$ in our experiments).

We perform a grid search using a held-out validation set to select the best performing hyperparameters for each model and dataset. The hyperparameter settings used for each model and dataset are described below.

**Symmetric low-rank DPP** (Gartrell et al., 2016). For this model, we use $K$ for the number of item feature dimensions for the symmetric component $\boldsymbol{V}$, and $\alpha$ for the regularization hyperparameter for $\boldsymbol{V}$. We use the following hyperparameter settings:

- Both Amazon datasets: $K = 30, \alpha = 0$.
- UK Retail dataset: $K = 100, \alpha = 1$.
- Instacart dataset: $K = 100, \alpha = 0.001$.
- Million Song dataset: $K = 150, \alpha = 0.0001$.

**Baseline NDPP** (Gartrell et al., 2019). For this model, to ensure consistency with the notation used in Gartrell et al. (2019), we use $D$ to denote the number of item feature dimensions for the symmetric component $\boldsymbol{V}$, and $D'$ to denote the number of item feature dimensions for the nonsymmetric components, $\boldsymbol{B}$ and $\boldsymbol{C}$. As described in Gartrell et al. (2019), $\alpha$ is the regularization hyperparameter for the $\boldsymbol{V}$, while $\beta$ and $\gamma$ are the regularization hyperparameters for $\boldsymbol{B}$ and $\boldsymbol{C}$, respectively. We use the following hyperparameter settings:

- Both Amazon datasets: $D = 30, \alpha = 0$.
- Amazon apparel dataset: $D' = 30$.
- Amazon three-category dataset: $D' = 100$.
- UK Retail dataset: $D = 100, D' = 20, \alpha = 1$.
- All datasets: $\beta = \gamma = 0$.

**Scalable NDPP**. As described in Section 3, we use $K$ to denote the number of item feature dimensions for the symmetric component $\boldsymbol{V}$ and the dimensionality of the nonsymmetric component $\boldsymbol{C}$. $\alpha$ is the regularization hyperparameter. We use the following hyperparameter settings:

- Amazon apparel dataset: $K = 30, \alpha = 0$.
- Amazon three-category dataset: $K = 100, \alpha = 1$.
- UK dataset: $K = 100, \alpha = 0.01$.

- Instacart dataset: $K = 100, \alpha = 0.001$.
- Million Song dataset: $K = 150, \alpha = 0.01$.

For all of the above model configurations we use a batch size of 200 during training, except for the scalable NDPPs trained on the Amazon apparel, Amazon three-category, Instacart, and Million Song datasets, where a batch size of 800 is used.

## C BENCHMARK ALGORITHMS FOR MAP INFERENCE

We test the following approximate algorithms for MAP inference:

**Greedy local search.** This algorithm starts from the output of greedy, $Y^G$, and replaces $i \in Y^G$ with $j \notin Y^G$ that gives the maximum improvement of the determinant, if such $i, j$ exist. Kathuria & Deshpande (2016) showed that running the search for such a swap $O(k^2 \log(k/\varepsilon))$ times with an accuracy parameter $\varepsilon$ gives a tight approximation guarantee for MAP inference for symmetric DPPs. We set the number of swaps to $\lfloor k^2 \log(10k) \rfloor$ for $\varepsilon = 0.1$ and use greedy local search as a baseline, since it is strictly an improvement on the greedy solution. The proposed greedy conditioning can be used for fast greedy local search. Specifically, for each $i \in Y^G$, Algorithm 1 can compute marginal improvements conditioned by $Y^G \setminus \{i\}$ in time $O(MKk)$, and thus its runtime can be $O(MKk^4 \log(k/\varepsilon))$. However, it is the slowest among all of our benchmark algorithms.

**Stochastic greedy.** This algorithm computes marginal gains of a few items chosen uniformly at random and selects the best among them. Mirzasoleiman et al. (2015) proved that $(M/k) \log(1/\varepsilon)$ samples are enough to guarantee an $(1 - 1/e - \varepsilon)$-approximation ratio for submodular functions (i.e., symmetric DPPs). We choose $\varepsilon = 0.1$ and set the number of samples to $\lfloor (M/k) \log(10) \rfloor$. Under this setting, the time complexity of stochastic greedy is $O(MKk^2 \log(1/\varepsilon))$, which is better than the naïve exact greedy algorithm. However, we note that it is worse than that of our efficient greedy implement (Algorithm 1). This is because the stochastic greedy uses different random samples for every iteration and this does not take advantage of the amortized computations in Lemma 2. In our experiments, we simply modify line 10 in Algorithm 1 for stochastic greedy ($\mathrm{argmax}$ is operated on a random subset of marginal gains), hence it can run in $O(MKk + (M/k) \log(1/\varepsilon))$ time. In practice, we observe that stochastic greedy is slightly slower than exact greedy due to the additional costs of the random sampling process.

**MCMC sampling.** We also compare inference algorithms with sampling from a nonsymmetric DPP. To the best of our knowledge, exact sampling of a non-Hermitian DPP was studied in Poulson (2019), which requires the Cholesky decomposition with $O(M^3)$ complexity. This is infeasible for a large $M$. To resolve this, Markov Chain Monte-Carlo (MCMC) based sampling is preferred (Li et al., 2016) for symmetric DPPs. In particular, we consider a Gibbs sampling for $k$-DPP, which begins with a random subset $Y$ with size $k$, and picks $i \in Y$ and $j \notin Y$ uniformly at random. Then, it swaps them with probability

$$\frac{\det(\boldsymbol{L}_{Y \cup \{j\} \setminus \{i\}})}{\det(\boldsymbol{L}_{Y \cup \{j\} \setminus \{i\}}) + \det(\boldsymbol{L}_Y)} \tag{11}$$

and repeat this process for several steps. Li et al. (2016) showed that $O(Nk \log(k/\varepsilon))$ swaps are enough to approximate the ground-truth distribution under symmetric DPPs. However, for a fair runtime comparison to Algorithm 1, we set the number of swaps to $\lfloor 3N/K \rfloor$.

## D COROLLARY OF THEOREM 2

Theorem 2 requires the technical condition $\sigma_{\min} > 1$, but in practice there is no particular evidence that this condition holds. While this condition can be achieved by multiplying $\boldsymbol{L}$ by a constant, this leads to a (potentially large) additive term in Eq. 10. Here, we provide Corollary 1 which excludes the $\sigma_{\min} > 1$ assumption from Theorem 2, and quantifies this additive term.

**Corollary 1.** *Consider a nonsymmetric low-rank DPP $\boldsymbol{L} = \boldsymbol{V}\boldsymbol{V}^\top + \boldsymbol{B}\boldsymbol{C}\boldsymbol{B}^\top$, where $\boldsymbol{V}, \boldsymbol{B}$ are of rank $K$, and $\boldsymbol{C} \in \mathbb{R}^{K \times K}$. Given a cardinality budget $k$, let $\sigma_{\min}$ and $\sigma_{\max}$ denote the smallest and*

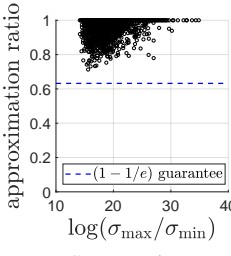
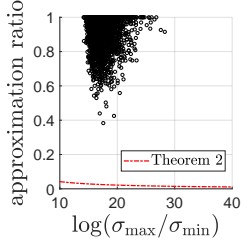
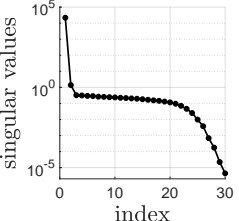

| (a) Symmetric DPP | (b) Nonsymmetric DPP | (c) Singular values |

Figure 2: Approximation ratios of greedy with respect to different values of $\log(\sigma_{\max}/\sigma_{\min})$ from Corollary 1 under (a) symmetric DPP and (b) nonsymmetric DPP. (c) The singular values of the kernels learned for the "Amazon: 3-category" dataset. We construct 10,000 random $P_0$ matrices $\boldsymbol{L} \in \mathbb{R}^{5 \times 5}$, with rank $K = 3$, whose singular values are from the learned kernels.

*largest singular values of $\boldsymbol{L}_Y$ for all $Y \subseteq [\![M]\!]$ and $|Y| \leq 2k$. Let $\kappa := \sigma_{\max}/\sigma_{\min}$. Then,*

$$\log \det(\boldsymbol{L}_{Y^G}) \geq \frac{4(1 - e^{-1/4})}{2 \log \kappa + 1} \log \det(\boldsymbol{L}_{Y^*}) - \left(1 - \frac{4(1 - e^{-1/4})}{2 \log \kappa + 1}\right) k \left(1 - \log \sigma_{\min}\right) \quad (12)$$

*where $Y^G$ is the output of Algorithm 1 and $Y^*$ is the optimal solution of MAP inference in Eq. 4.*

The proof of Corollary 1 is provided in Appendix F.5. Note that instead of $\log(\sigma_{\max})/\log(\sigma_{\min})$, Corollary 1 has a $\log(\sigma_{\max}/\sigma_{\min})$ term in the denominator.

## E    PERFORMANCE GUARANTEE FOR GREEDY MAP INFERENCE

The matrices learned on real datasets are too large to compute the exact MAP solution, but we can compute exact MAP for small matrices. In this section, we explore the performance of the greedy algorithm studied in Theorem 2 for $5 \times 5$ synthetic kernel matrices. More formally, we first pick $K = 3$ singular values $s_1, s_2, s_3$ from a kernel learned for the "Amazon: 3-category" dataset (a plot of these singular values can be seen in Fig. 2(c)) and generate $\boldsymbol{L} = \boldsymbol{V}_1 \mathtt{diag}([s_1, s_2, s_3]) \boldsymbol{V}_2^\top$, where $\boldsymbol{V}_1, \boldsymbol{V}_2 \in \mathbb{R}^{5 \times 3}$ are random orthonormal matrices. To ensure that $\boldsymbol{L}$ is a $P_0$ matrix, we repeatedly sample $\boldsymbol{V}_1, \boldsymbol{V}_2$ until all principal minors of $\boldsymbol{L}$ are nonnegative. We also evaluate the performance of the symmetric DPP, where the kernel matrices are generated similarly to the NDPP, except we set $\boldsymbol{V}_1 = \boldsymbol{V}_2$. We set $k = 3$ and generate 10,000 random kernels for both symmetric DPPs and NDPPs.

The results for symmetric and nonsymmetric DPPs are shown in Fig. 2(a) and Fig. 2(b), respectively. We plot the approximation ratio of Algorithm 1, i.e., $\log \det(\boldsymbol{L}_{Y^G})/\log \det(\boldsymbol{L}_{Y^*})$, with respect to $\log(\sigma_{\max}/\sigma_{\min})$, from Corollary 1. We observe that the greedy algorithm for both often shows approximation ratios close to 1. However, the worst-case ratio for NDPPs is worse than that of symmetric DPPs; $\log \det(\boldsymbol{L}_Y)$ for $\boldsymbol{L} \in P_0^+$ is non-submodular, and the greedy algorithm with a nonsubmodular function does not have as tight of a worst-case bound as in the symmetric case.

## F PROOFS

### F.1 PROOF OF LEMMA 1

**Lemma 1.** *Let $\boldsymbol{A} \in \mathbb{R}^{M \times M}$ be a skew-symmetric matrix with rank $\ell \leq M$. Then, there exist $\boldsymbol{B} \in \mathbb{R}^{M \times \ell}$ and positive numbers $\lambda_1, \ldots, \lambda_{\lfloor \ell/2 \rfloor}$, such that $\boldsymbol{A} = \boldsymbol{BCB}^\top$, where $\boldsymbol{C} \in \mathbb{R}^{\ell \times \ell}$ is the block-diagonal matrix with $\lfloor \ell/2 \rfloor$ diagonal blocks of size 2 given by $\Sigma_i$, $i = 1, \ldots, \lfloor \ell/2 \rfloor$ and zero elsewhere.*

*Proof.* First, we note that rank of a nonsingular skew-symmetric matrix is always even, because all of its eigenvalues are purely imaginary and come in conjugate pairs. There exists some orthogonal matrix $\mathbf{P} \in \mathbb{R}^{M \times M}$ and

$$
\boldsymbol{\Sigma} = \begin{pmatrix}
0 & \lambda_1 & & & & & & & \\
-\lambda_1 & 0 & & & & & & & \\
& & 0 & \lambda_2 & & & \mathbf{0} & & \\
& & -\lambda_2 & 0 & & & & & \\
& & & & \ddots & & & & \\
& & & & & 0 & \lambda_{\ell/2} & & \\
& \mathbf{0} & & & & -\lambda_{\ell/2} & 0 & & \\
& & & & & & & 0 & \\
& & & & & & & & \ddots \\
& & & & & & & & & 0
\end{pmatrix}
\tag{13}
$$

such that $\boldsymbol{A} = \mathbf{P}\boldsymbol{\Sigma}\mathbf{P}^\top$ (see, e.g.,(Thompson, 1988, Proposition 2.1)).

Let $\boldsymbol{C}$ be the $\ell \times \ell$ supmatrix of $\boldsymbol{\Sigma}$ obtained by keeping its first $\ell$ rows and columns and let $\mathbf{Q} = \begin{pmatrix} \boldsymbol{I}_\ell \\ 0 \end{pmatrix}$, where $\boldsymbol{I}_\ell$ is the $\ell \times \ell$ identity matrix. Then, $\boldsymbol{\Sigma} = \boldsymbol{QCQ}^\top$ and one can write $\boldsymbol{A} = \boldsymbol{PQCQ}^\top \boldsymbol{P}^\top$. Setting $\boldsymbol{B} = \boldsymbol{PQ}$ proves the lemma. $\qquad\square$

### F.2 PROOF OF THEOREM 1

**Theorem 1.** *Given an NDPP with kernel $\boldsymbol{L} = \boldsymbol{VV}^\top + \boldsymbol{BCB}^\top$, parameterized by $\boldsymbol{V}$ of rank $K$, $\boldsymbol{B}$ of rank $K$, and a $K \times K$ matrix $\boldsymbol{C}$, we can compute the regularized log-likelihood (Eq. 2) and its gradient in $O(MK^2 + K^3 + nK'^3)$ time, where $K'$ is the size of the largest of the $n$ training subsets.*

*Proof.* We first show that the log-likelihood can be computed in time linear in $M$. Using the matrix determinant lemma, one can easily verify that the DPP normalization term can be computed as

$$
\det(\boldsymbol{I} + \boldsymbol{L}) = \det\left(\boldsymbol{I} + (\boldsymbol{V} \quad \boldsymbol{BC}) \begin{pmatrix} \boldsymbol{V}^\top \\ \boldsymbol{B}^\top \end{pmatrix}\right) = \det\left(\boldsymbol{I}_{2K} + \begin{pmatrix} \boldsymbol{V}^\top \\ \boldsymbol{B}^\top \end{pmatrix} (\boldsymbol{V} \quad \boldsymbol{BC})\right)
\tag{14}
$$

where $\boldsymbol{I}_{2K}$ is the identity matrix with dimension $2K$. As Eq. 14 requires a matrix-multiplication between $(2K) \times M$ matrices and the determinant of $(2K) \times (2K)$ matrices, this allows us to transform a $O(M^3)$ operation into an $O(MK^2 + K^3)$ one.

Having established that the normalization term in the likelihood can be computed in $O(MK^2 + K^3)$ time, we proceed with characterizing the complexity of the other terms in the likelihood. The first term in Eq. 2 consists of determinants of size $|Y_i|$. Assuming that these never exceed size $K'$, each can be computed in at most $O(K'^3)$ time. The regularization term is a simple sum of norms that can be computed in $O(MK)$ time. Therefore, the full regularized log-likelihood can be computed in $O(MK^2 + K^3 + nK'^3)$ time.

To prove that the gradient of the log-likelihood can be computed in time linear in $M$, we begin by showing that the logarithm of DPP normalization term can be factorized as follows:

$$Z = \log \det(\boldsymbol{I} + \boldsymbol{L}) \tag{15}$$

$$= \log \det \left( \boldsymbol{I}_{2K} + \begin{pmatrix} \boldsymbol{V}^\top \\ \boldsymbol{B}^\top \end{pmatrix} (\boldsymbol{V} \quad \boldsymbol{B}) \begin{pmatrix} \boldsymbol{I}_K & \boldsymbol{0} \\ \boldsymbol{0} & \boldsymbol{C} \end{pmatrix} \right) \tag{16}$$

$$= \log \det \left( \begin{pmatrix} \boldsymbol{I}_K & \boldsymbol{0} \\ \boldsymbol{0} & \boldsymbol{C}^{-1} \end{pmatrix} + \begin{pmatrix} \boldsymbol{V}^\top \\ \boldsymbol{B}^\top \end{pmatrix} (\boldsymbol{V} \quad \boldsymbol{B}) \right) + \log \det \begin{pmatrix} \boldsymbol{I}_K & \boldsymbol{0} \\ \boldsymbol{0} & \boldsymbol{C} \end{pmatrix} \tag{17}$$

$$= \log \det \begin{pmatrix} \boldsymbol{I}_K + \boldsymbol{V}^\top \boldsymbol{V} & \boldsymbol{V}^\top \boldsymbol{B} \\ \boldsymbol{B}^\top \boldsymbol{V} & \boldsymbol{C}^{-1} + \boldsymbol{B}^\top \boldsymbol{B} \end{pmatrix} + \log \det(\boldsymbol{C}) \tag{18}$$

$$= \log \det \left( \boldsymbol{I}_K + \boldsymbol{V}^\top \boldsymbol{V} \right) + \log \det \left( \boldsymbol{C}^{-1} + \boldsymbol{B}^\top (\boldsymbol{I} - \boldsymbol{V}(\boldsymbol{I}_K + \boldsymbol{V}^\top \boldsymbol{V})^{-1} \boldsymbol{V}^\top) \boldsymbol{B} \right) + \log \det(\boldsymbol{C}) \tag{19}$$

where Eq. 17 follows from the determinant commutativity (i.e., $\det(\boldsymbol{AB}) = \det(\boldsymbol{A}) \det(\boldsymbol{B})$) and Eq. 18 and Eq. 19 come from the Schur's determinant identity[†]. For simplicity, we write $\boldsymbol{X} = \boldsymbol{I} - \boldsymbol{V}(\boldsymbol{I}_K + \boldsymbol{V}^\top \boldsymbol{V})^{-1} \boldsymbol{V}^\top$ and $(\boldsymbol{C}^{-1})^\top = \boldsymbol{C}^{-\top}$, and note that $\boldsymbol{X}$ depends only on $\boldsymbol{V}$.

The gradient of $Z$ has three parts: $\nabla Z = (\nabla_{\boldsymbol{V}} Z, \nabla_{\boldsymbol{B}} Z, \nabla_{\boldsymbol{C}} Z)$ where each can be computed as

$$\nabla_{\boldsymbol{V}} Z = \nabla_{\boldsymbol{V}} \log \det(\boldsymbol{I}_K + \boldsymbol{V}^\top \boldsymbol{V}) + \nabla_{\boldsymbol{V}} \log \det(\boldsymbol{C}^{-1} + \boldsymbol{B}^\top \boldsymbol{X} \boldsymbol{B}) \tag{20}$$

$$= 2\boldsymbol{V}(\boldsymbol{I}_K + \boldsymbol{V}^\top \boldsymbol{V})^{-1}$$
$$- \boldsymbol{X}\boldsymbol{B}((\boldsymbol{C}^{-1} + \boldsymbol{B}^\top \boldsymbol{X} \boldsymbol{B})^{-1} + (\boldsymbol{C}^{-\top} + \boldsymbol{B}^\top \boldsymbol{X} \boldsymbol{B})^{-1}) \boldsymbol{B}^\top \boldsymbol{X} \boldsymbol{V} \tag{21}$$

$$\nabla_{\boldsymbol{B}} Z = \nabla_{\boldsymbol{B}} \log \det(\boldsymbol{C}^{-1} + \boldsymbol{B}^\top \boldsymbol{X} \boldsymbol{B}) \tag{22}$$

$$= \boldsymbol{X}\boldsymbol{B} \left( (\boldsymbol{C}^{-1} + \boldsymbol{B}^\top \boldsymbol{X} \boldsymbol{B})^{-1} + (\boldsymbol{C}^{-\top} + \boldsymbol{B}^\top \boldsymbol{X} \boldsymbol{B})^{-1} \right) \tag{23}$$

$$\nabla_{\boldsymbol{C}} Z = \nabla_{\boldsymbol{C}} \log \det(\boldsymbol{C}) + \nabla_{\boldsymbol{C}} \log \det(\boldsymbol{C}^{-1} + \boldsymbol{B}^\top \boldsymbol{X} \boldsymbol{B}) \tag{24}$$

$$= \boldsymbol{C}^{-\top} - \boldsymbol{C}^{-\top}(\boldsymbol{C}^{-1} + \boldsymbol{B}^\top \boldsymbol{X} \boldsymbol{B})^{-\top} \boldsymbol{C}^{-\top} \tag{25}$$

Observe that $\boldsymbol{X}$ combines a $M \times M$ identity matrix with $M \times K$ matrices, hence multiplying it with a $M \times K$ matrix (e.g., $\boldsymbol{XV}$ or $\boldsymbol{XB}$) can be computed in $O(MK^2)$ time. Since each of the remaining matrix inverses in Eq. 21, Eq. 23, and Eq. 25 involve a $K \times K$ matrix inverse, with a cost of $O(K^3)$ operations, we have a net computational cost of $O(MK^2 + K^3)$ for computing $\nabla \log \det(\boldsymbol{I} + \boldsymbol{L})$.

The gradient of the first term in Eq. 2 involves computing gradients of determinants of size at most $K'$, which results in size $K'$ matrix inverses, since for a matrix $\boldsymbol{A}$, $\frac{\partial}{\partial A_{ij}}(\log \det(\boldsymbol{A})) = (\boldsymbol{A}^{-1})_{ij}^\top$. Each of these inverses can be computed in $O(K'^3)$ time. The gradient of the simple sum-of-norms regularization term can be computed in $O(MK)$ time. Therefore, combining these results with the results above for the complexity of the gradient of the normalization term, we have the following overall complexity of the gradient for the full log-likelihood: $O(MK^2 + K^3 + nK'^3)$. $\qquad \square$

### F.3 PROOF OF LEMMA 2

**Lemma 2.** *Given $\boldsymbol{B} \in \mathbb{R}^{M \times K}$, $\boldsymbol{C} \in \mathbb{R}^{K \times K}$, and $Y = \{a_1, \ldots, a_k\} \subseteq \llbracket M \rrbracket$, let $\boldsymbol{b}_i \in \mathbb{R}^{1 \times K}$ be the $i$-th row in $\boldsymbol{B}$ and $\boldsymbol{B}_Y \in \mathbb{R}^{|Y| \times K}$ be a matrix containing rows in $\boldsymbol{B}$ indexed by $Y$. Then, it holds that*

$$\boldsymbol{B}_Y^\top (\boldsymbol{B}_Y \boldsymbol{C} \boldsymbol{B}_Y^\top)^{-1} \boldsymbol{B}_Y = \sum_{j=1}^k \boldsymbol{p}_j^\top \boldsymbol{q}_j, \tag{6}$$

*where row vectors $\boldsymbol{p}_j, \boldsymbol{q}_j \in \mathbb{R}^{1 \times K}$ for $j = 1, \ldots, k$ satisfy $\boldsymbol{p}_1 = \boldsymbol{b}_{a_1}/(\boldsymbol{b}_{a_1} \boldsymbol{C} \boldsymbol{b}_{a_1}^\top)$, $\boldsymbol{q}_1 = \boldsymbol{b}_{a_1}$, and*

$$\boldsymbol{p}_{j+1} = \frac{\boldsymbol{b}_{a_j} - \boldsymbol{b}_{a_j} \boldsymbol{C}^\top \sum_{i=1}^j \boldsymbol{q}_i^\top \boldsymbol{p}_i}{\boldsymbol{b}_{a_j} \boldsymbol{C}(\boldsymbol{b}_{a_j} - \boldsymbol{b}_{a_j} \boldsymbol{C}^\top \sum_{i=1}^j \boldsymbol{q}_i^\top \boldsymbol{p}_i)^\top}, \qquad \boldsymbol{q}_{j+1} = \boldsymbol{b}_{a_j} - \boldsymbol{b}_{a_j} \boldsymbol{C} \sum_{i=1}^j \boldsymbol{p}_i^\top \boldsymbol{q}_i. \tag{7}$$

---

[†]$\det \begin{pmatrix} \boldsymbol{A} & \boldsymbol{B} \\ \boldsymbol{C} & \boldsymbol{D} \end{pmatrix} = \det(\boldsymbol{A}) \det(\boldsymbol{D} - \boldsymbol{C} \boldsymbol{A}^{-1} \boldsymbol{B}).$

*Proof.* We prove by induction on $k$. When $k = 1$, the result is trivial because

$$\boldsymbol{B}_Y^\top (\boldsymbol{B}_Y \boldsymbol{C} \boldsymbol{B}_Y^\top)^{-1} \boldsymbol{B}_Y = \boldsymbol{b}_{a_1}^\top (\boldsymbol{b}_{a_1} \boldsymbol{C} \boldsymbol{b}_{a_1}^\top)^{-1} \boldsymbol{b}_{a_1} = \boldsymbol{p}_1^\top \boldsymbol{q}_1. \tag{26}$$

Now we assume that the statement holds for $k - 1$. Let $Y := \{a_1, \ldots, a_{k-1}\}$ and $a := a_k$. From the inductive hypothesis, it holds

$$\boldsymbol{B}_Y^\top (\boldsymbol{B}_Y \boldsymbol{C} \boldsymbol{B}_Y^\top)^{-1} \boldsymbol{B}_Y = \sum_{j=1}^{k-1} \boldsymbol{p}_j^\top \boldsymbol{q}_j. \tag{27}$$

Now we write

$$\boldsymbol{B}_{Y \cup \{a\}}^\top \left( \boldsymbol{B}_{Y \cup \{a\}} \boldsymbol{C} \boldsymbol{B}_{Y \cup \{a\}}^\top \right)^{-1} \boldsymbol{B}_{Y \cup \{a\}} \tag{28}$$

$$= \boldsymbol{B}_{Y \cup \{a\}}^\top \left( \begin{pmatrix} \boldsymbol{B}_Y \\ \boldsymbol{b}_a \end{pmatrix} \boldsymbol{C} \begin{pmatrix} \boldsymbol{B}_Y^\top & \boldsymbol{b}_a^\top \end{pmatrix} \right)^{-1} \boldsymbol{B}_{Y \cup \{a\}} \tag{29}$$

$$= \begin{pmatrix} \boldsymbol{B}_Y^\top & \boldsymbol{b}_a^\top \end{pmatrix} \begin{pmatrix} \boldsymbol{B}_Y \boldsymbol{C} \boldsymbol{B}_Y^\top & \boldsymbol{B}_Y \boldsymbol{C} \boldsymbol{b}_a^\top \\ \boldsymbol{b}_a \boldsymbol{C} \boldsymbol{B}_Y^\top & \boldsymbol{b}_a \boldsymbol{C} \boldsymbol{b}_a^\top \end{pmatrix}^{-1} \begin{pmatrix} \boldsymbol{B}_Y \\ \boldsymbol{b}_a \end{pmatrix}. \tag{30}$$

To handle the inverse matrix we employ the Schur complement, which yields

$$\begin{pmatrix} \boldsymbol{X} & \boldsymbol{y} \\ \boldsymbol{z} & w \end{pmatrix}^{-1} = \begin{pmatrix} \boldsymbol{X}^{-1} & \boldsymbol{0} \\ \boldsymbol{0} & 0 \end{pmatrix} + \frac{1}{w - \boldsymbol{z} X^{-1} \boldsymbol{y}} \begin{pmatrix} \boldsymbol{X}^{-1} \boldsymbol{y} \boldsymbol{z} \boldsymbol{X}^{-1} & -\boldsymbol{X}^{-1} \boldsymbol{y} \\ -\boldsymbol{z} \boldsymbol{X}^{-1} & 1 \end{pmatrix} \tag{31}$$

for any non-singular square matrix $\boldsymbol{X} \in \mathbb{R}^{k \times k}$, column vector $\boldsymbol{y} \in \mathbb{R}^k$ and row vector $\boldsymbol{z} \in \mathbb{R}^{1 \times k}$, unless $(w - \boldsymbol{z} X^{-1} \boldsymbol{y}) = 0$. Applying this, we have

$$\begin{pmatrix} \boldsymbol{B}_Y \boldsymbol{C} \boldsymbol{B}_Y^\top & \boldsymbol{B}_Y \boldsymbol{C} \boldsymbol{b}_a^\top \\ \boldsymbol{b}_a \boldsymbol{C} \boldsymbol{B}_Y^\top & \boldsymbol{b}_a \boldsymbol{C} \boldsymbol{b}_a^\top \end{pmatrix}^{-1} = \begin{pmatrix} (\boldsymbol{B}_Y \boldsymbol{C} \boldsymbol{B}_Y^\top)^{-1} & \boldsymbol{0} \\ \boldsymbol{0} & 0 \end{pmatrix} + \frac{1}{\boldsymbol{b}_a \boldsymbol{C} \boldsymbol{b}_a^\top - \boldsymbol{b}_a \boldsymbol{C} \boldsymbol{B}_Y^\top (\boldsymbol{B}_Y \boldsymbol{C} \boldsymbol{B}_Y^\top)^{-1} \boldsymbol{B}_Y \boldsymbol{C} \boldsymbol{b}_a^\top}$$
$$\begin{pmatrix} (\boldsymbol{B}_Y \boldsymbol{C} \boldsymbol{B}_Y^\top)^{-1} \boldsymbol{B}_Y \boldsymbol{C} \boldsymbol{b}_a^\top \boldsymbol{b}_a \boldsymbol{C} \boldsymbol{B}_Y^\top (\boldsymbol{B}_Y \boldsymbol{C} \boldsymbol{B}_Y^\top)^{-1} & -(\boldsymbol{B}_Y \boldsymbol{C} \boldsymbol{B}_Y^\top)^{-1} \boldsymbol{B}_Y \boldsymbol{C} \boldsymbol{b}_a^\top \\ -\boldsymbol{b}_a \boldsymbol{C} \boldsymbol{B}_Y^\top (\boldsymbol{B}_Y \boldsymbol{C} \boldsymbol{B}_Y^\top)^{-1} & 1. \end{pmatrix} \tag{32}$$

Substituting Eq. 32 into Eq. 30, we obtain

$$\boldsymbol{B}_{Y \cup \{a\}}^\top \left( \boldsymbol{B}_{Y \cup \{a\}} \boldsymbol{C} \boldsymbol{B}_{Y \cup \{a\}}^\top \right)^{-1} \boldsymbol{B}_{Y \cup \{a\}} \tag{33}$$

$$= \boldsymbol{B}_Y^\top \left( \boldsymbol{B}_Y \boldsymbol{C} \boldsymbol{B}_Y^\top \right)^{-1} \boldsymbol{B}_Y + \frac{\left( \boldsymbol{b}_a^\top - \boldsymbol{B}_Y^\top (\boldsymbol{B}_Y \boldsymbol{C} \boldsymbol{B}_Y^\top)^{-1} \boldsymbol{B}_Y \boldsymbol{C} \boldsymbol{b}_a^\top \right) \left( \boldsymbol{b}_a - \boldsymbol{b}_a \boldsymbol{C} \boldsymbol{B}_Y^\top (\boldsymbol{B}_Y \boldsymbol{C} \boldsymbol{B}_Y^\top)^{-1} \boldsymbol{B}_Y \right)}{\boldsymbol{b}_a \boldsymbol{C} \left( \boldsymbol{b}_a^\top - \boldsymbol{B}_Y^\top (\boldsymbol{B}_Y \boldsymbol{C} \boldsymbol{B}_Y^\top)^{-1} \boldsymbol{B}_Y \boldsymbol{C} \boldsymbol{b}_a^\top \right)} \tag{34}$$

$$= \sum_{j=1}^{k-1} \boldsymbol{p}_j^\top \boldsymbol{q}_j + \frac{\left( \boldsymbol{b}_a^\top - \sum_{j=1}^{k-1} \boldsymbol{p}_j^\top \boldsymbol{q}_j \boldsymbol{C} \boldsymbol{b}_a^\top \right) \left( \boldsymbol{b}_a - \boldsymbol{b}_a \boldsymbol{C} \sum_{j=1}^{k-1} \boldsymbol{p}_j^\top \boldsymbol{q}_j \right)}{\boldsymbol{b}_a \boldsymbol{C} \left( \boldsymbol{b}_a^\top - \sum_{j=1}^{k-1} \boldsymbol{p}_j^\top \boldsymbol{q}_j \boldsymbol{C} \boldsymbol{b}_a^\top \right)} \tag{35}$$

$$= \sum_{j=1}^{k-1} \boldsymbol{p}_j^\top \boldsymbol{q}_j + \boldsymbol{p}_k^\top \boldsymbol{q}_k \tag{36}$$

where the third line holds from the inductive hypothesis Eq. 27 and the last line holds from the definition of $\boldsymbol{p}_k, \boldsymbol{q}_k \in \mathbb{R}^{1 \times K}$. $\qquad\square$

### F.4 PROOF OF THEOREM 2

**Theorem 2.** *Consider a nonsymmetric low-rank DPP $\boldsymbol{L} = \boldsymbol{V} \boldsymbol{V}^\top + \boldsymbol{B} \boldsymbol{C} \boldsymbol{B}^\top$, where $\boldsymbol{V}, \boldsymbol{B}$ are of rank $K$, and $\boldsymbol{C} \in \mathbb{R}^{K \times K}$. Given a cardinality budget $k$, let $\sigma_{\min}$ and $\sigma_{\max}$ denote the smallest and largest singular values of $\boldsymbol{L}_Y$ for all $Y \subseteq [\![M]\!]$ and $|Y| \leq 2k$. Assume that $\sigma_{\min} > 1$. Then,*

$$\log \det(\boldsymbol{L}_{Y^G}) \geq \frac{4(1 - e^{-1/4})}{2(\log \sigma_{\max} / \log \sigma_{\min}) - 1} \log \det(\boldsymbol{L}_{Y^*}) \tag{10}$$

*where $Y^G$ is the output of Algorithm 1 and $Y^*$ is the optimal solution of MAP inference in Eq. 4.*

*Proof.* The proof of Theorem 2 relies on an approximation guarantee for nonsubmodular greedy maximization (Bian et al., 2017, Theorem 1). We introduce their result below.

**Theorem 3** ((Bian et al., 2017, Theorem 1)). *Consider a set function $f$ defined on all subsets of $\{1, \ldots, M\} = \llbracket M \rrbracket$ is monotone nondecreasing and nonnegative, i.e., $0 \leq f(Y) \leq f(X)$ for $\forall Y \subseteq X \subseteq \llbracket M \rrbracket$. Given a cardinality budget $k \geq 1$, let $Y^*$ be the optimal solution of $\max_{|Y|=k} f(Y)$ and $Y^0 = \emptyset$, $Y^t := \{a_1, \ldots, a_t\}, t = 1, \ldots, k$ be the successive chosen by the greedy algorithm with budget $k$. Denote $\gamma$ be the largest scalar such that*

$$\sum_{i \in X \setminus Y^t} (f(Y^t \cup \{i\}) - f(Y^t)) \geq \gamma(f(X \cup Y^t) - f(Y^t)), \tag{37}$$

*for $\forall X \subseteq \llbracket M \rrbracket, |X| = k$ and $t = 0, \ldots, k-1$, and $\alpha$ be the smallest scalar such that*

$$f(Y^{t-1} \cup \{i\} \cup X) - f(Y^{t-1} \cup X) \geq (1-\alpha) \left( f(Y^{t-1} \cup \{i\}) - f(Y^{t-1}) \right). \tag{38}$$

*for $\forall X \subseteq \llbracket M \rrbracket, |X| = k$ and $i \in Y^{k-1} \setminus X$. Then, it holds that*

$$f(Y^k) \geq \frac{1}{\alpha} \left(1 - e^{-\alpha \gamma}\right) f(Y^*). \tag{39}$$

In order to apply this result for MAP inference of NDPPs, the objective should be monotone nondecreasing and nonnegative. We first show that $\sigma_{\min} > 1$ is a sufficient condition for both monotonicity and nonnegativity.

**Lemma 3.** *Given a $P_0$ matrix $\boldsymbol{L} \in \mathbb{R}^{M \times M}$ and the budget $k \geq 0$, a set function $f(Y) = \log \det(\boldsymbol{L}_Y)$ defined on $Y \subseteq \llbracket M \rrbracket$ is monotone nondecreasing and nonnegative for $|Y| \leq k$ when $\sigma_{\min} > 1$.*

The proof of Lemma 3 is provided in Appendix F.6. Next, we aim to find proper bounds on $\alpha$ and $\gamma$. To resolve this, we provide the following upper and lower bounds of the marginal gain for $f(Y) = \log \det(\boldsymbol{L}_Y)$.

**Lemma 4.** *Let $f(Y) = \log \det(\boldsymbol{L}_Y)$ and assume that $\sigma_{\min} > 1$. Then, for $Y \subseteq \llbracket M \rrbracket, |Y| < 2k$ and $i \notin Y$, it holds that*

$$f(Y \cup \{i\}) - f(Y) \geq \log \sigma_{\min}, \tag{40}$$
$$f(Y \cup \{i\}) - f(Y) \leq 2 \log \sigma_{\max} - \log \sigma_{\min} \tag{41}$$

*where $\sigma_{\min}$ and $\sigma_{\max}$ are the smallest and largest singular values of $\boldsymbol{L}_Y$ for all $Y \subseteq \llbracket M \rrbracket, |Y| \leq 2k$.*

The proof of Lemma 4 is provided in Appendix F.7. To bound $\gamma$, we consider $X \subseteq \llbracket M \rrbracket, |X| = k$ and denote $X \setminus Y^t = \{x_1, \ldots, x_r\} \neq \emptyset$. Then,

$$\sum_{i \in X \setminus Y^t} (f(Y^t \cup \{i\}) - f(Y)) = \sum_{j=1}^{r} f(Y^t \cup \{x_r\}) - f(Y^t) \geq r \log \sigma_{\min} \tag{42}$$

where the last inequality comes from Eq. 40. Similarly, we get

$$f(X \cup Y^t) - f(Y^t) = \sum_{j=1}^{r} f(\{x_1, \ldots, x_j\} \cup Y^t) - f(\{x_1, \ldots, x_{j-1}\} \cup Y^t) \tag{43}$$
$$\leq r(2 \log \sigma_{\max} - \log \sigma_{\min}) \tag{44}$$

where the last inequality comes from Eq. 41. Combining Eq. 42 to Eq. 44, we obtain that

$$\frac{\sum_{i \in X \setminus Y^t} f(Y^t \cup \{i\}) - f(Y^t)}{f(X \cup Y^t) - f(Y^t)} \geq \frac{\log \sigma_{\min}}{2 \log \sigma_{\max} - \log \sigma_{\min}} \tag{45}$$

which allows us to choose $\gamma = \left(2 \frac{\log \sigma_{\max}}{\log \sigma_{\min}} - 1\right)^{-1}$.

To bound $\alpha$, we similarly use Lemma 4 to obtain

$$\frac{f(X \cup Y^{t-1} \cup \{i\}) - f(X \cup Y^{t-1})}{f(Y^{t-1} \cup \{i\}) - f(Y^{t-1})} \geq \frac{\log \sigma_{\min}}{2 \log \sigma_{\max} - \log \sigma_{\min}} \tag{46}$$

and we can choose $\alpha = 1 - \frac{\log \sigma_{\min}}{2 \log \sigma_{\max} - \log \sigma_{\min}} = \frac{2(\log \sigma_{\max} - \log \sigma_{\min})}{2 \log \sigma_{\max} - \log \sigma_{\min}}$.

Now let $\kappa = \frac{\log \sigma_{\max}}{\log \sigma_{\min}}$. Then $\gamma = \frac{1}{2\kappa - 1}$ and $\alpha = \frac{2(\kappa - 1)}{2\kappa - 1}$. Putting $\gamma$ and $\alpha$ into Eq. 39, we have

$$\frac{1}{\alpha}(1 - e^{-\alpha\gamma}) \geq \frac{2\kappa - 1}{2(\kappa - 1)} \left(1 - e^{-\frac{2(\kappa - 1)}{(2\kappa - 1)^2}}\right) \tag{47}$$

$$\geq \frac{2\kappa - 1}{2(\kappa - 1)} \, 4\exp(-1/4) \frac{2(\kappa - 1)}{(2\kappa - 1)^2} \tag{48}$$

$$= \frac{4\exp(-1/4)}{2\kappa - 1} \tag{49}$$

where the second inequality holds from the fact that $\max_{\kappa \geq 1} \frac{2(\kappa - 1)}{(2\kappa - 1)^2} = \frac{1}{4}$ and $1 - e^{-x} \geq 4\exp(-1/4)x$ for $x \in [0, 1/4]$. □

## F.5 PROOF OF COROLLARY 1

**Corollary 1.** *Consider a nonsymmetric low-rank DPP $\boldsymbol{L} = \boldsymbol{V}\boldsymbol{V}^\top + \boldsymbol{B}\boldsymbol{C}\boldsymbol{B}^\top$, where $\boldsymbol{V}, \boldsymbol{B}$ are of rank $K$, and $\boldsymbol{C} \in \mathbb{R}^{K \times K}$. Given a cardinality budget $k$, let $\sigma_{\min}$ and $\sigma_{\max}$ denote the smallest and largest singular values of $\boldsymbol{L}_Y$ for all $Y \subseteq [\![M]\!]$ and $|Y| \leq 2k$. Let $\kappa := \sigma_{\max}/\sigma_{\min}$. Then,*

$$\log \det(\boldsymbol{L}_{Y^G}) \geq \frac{4(1 - e^{-1/4})}{2\log\kappa + 1} \log \det(\boldsymbol{L}_{Y^*}) - \left(1 - \frac{4(1 - e^{-1/4})}{2\log\kappa + 1}\right) k \, (1 - \log \sigma_{\min}) \tag{12}$$

*where $Y^G$ is the output of Algorithm 1 and $Y^*$ is the optimal solution of MAP inference in Eq. 4.*

*Proof.* Now consider $\boldsymbol{L}' = (\frac{e}{\sigma_{\min}})\boldsymbol{L}$ where $e$ is the exponential constant. Then, $\sigma'_{\min} = \sigma_{\min}(\frac{e}{\sigma_{\min}}) = e$ and $\sigma'_{\max} = \sigma_{\max}(\frac{e}{\sigma_{\min}})$. Using the fact that $\log \det(\boldsymbol{L}'_Y) = \log \det(\boldsymbol{L}_Y) - |Y|\log \sigma_{\min}$, we obtain the result. □

## F.6 PROOF OF LEMMA 3

Before stating the proof, we introduce interlacing properties of singular values.

**Theorem 4** (Interlacing Inequality for Singular Values, (Thompson, 1972, Theorem 1)). *Consider a real matrix $\boldsymbol{A} \in \mathbb{R}^{M \times N}$ with singular values $\sigma_1 \geq \sigma_2 \geq \cdots \geq \sigma_{\min(M,N)}$ and its supmatrix $\boldsymbol{B} \in \mathbb{R}^{P \times Q}$ with singular values $\beta_1 \geq \beta_2 \geq \cdots \geq \beta_{\min(P,Q)}$. Then, the singular values have the following interlacing properties:*

$$\sigma_i \geq \beta_i, \qquad\qquad for \; i = 1, \ldots, \min(P, Q), \tag{50}$$

$$\beta_i \geq \sigma_{i+M-P+N-Q}, \qquad for \; i = 1, \ldots, \min(P + Q - M, P + Q - N). \tag{51}$$

*Note that when $M = N$ and $P = Q = N - 1$, it holds that $\beta_i \geq \sigma_{i+2}$ for $i = 1, \ldots, N - 2$.*

We are now ready to prove Lemma 3.

**Lemma 3.** *Given a $P_0$ matrix $\boldsymbol{L} \in \mathbb{R}^{M \times M}$ and the budget $k \geq 0$, a set function $f(Y) = \log \det(\boldsymbol{L}_Y)$ defined on $Y \subseteq [\![M]\!]$ is monotone nondecreasing and nonnegative for $|Y| \leq k$ when $\sigma_{\min} > 1$.*

*Proof.* Since $\boldsymbol{L} \in P_0$, all of its principal submatrices are also in $P_0$. By the definition of a $P_0$ matrix, it holds that

$$|\det(\boldsymbol{L}_Y)| = \det(\boldsymbol{L}_Y) = \prod_i \sigma_i(\boldsymbol{L}_Y) \tag{52}$$

where $\sigma_i(\boldsymbol{L}_Y)$ for $i \in [\![|Y|]\!]$ are singular values of $\boldsymbol{L}_Y$. Then, $F(Y) = \sum_i \log(\sigma_i(\boldsymbol{L}_Y))$ is nonnegative for all $Y$ such that $|Y| \leq K$ due to $\sigma_i(\boldsymbol{L}_Y) \geq \sigma_{\min} > 1$. Similarly, we have $F(Y \cup \{a\}) - F(Y) = \sum_{i=1}^{|Y|+1} \log \sigma_i(\boldsymbol{L}_{Y \cup \{a\}}) - \sum_{i=1}^{|Y|} \log \sigma_i(\boldsymbol{L}_Y) \geq \log \sigma_{\min} > 0$ from Eq. 50. □

### F.7 PROOF OF LEMMA 4

**Lemma 4.** *Let $f(Y) = \log\det(\boldsymbol{L}_Y)$ and assume that $\sigma_{\min} > 1$. Then, for $Y \subseteq [\![M]\!], |Y| < 2k$ and $i \notin Y$, it holds that*

$$f(Y \cup \{i\}) - f(Y) \geq \log\sigma_{\min}, \tag{40}$$
$$f(Y \cup \{i\}) - f(Y) \leq 2\log\sigma_{\max} - \log\sigma_{\min} \tag{41}$$

*where $\sigma_{\min}$ and $\sigma_{\max}$ are the smallest and largest singular values of $\boldsymbol{L}_Y$ for all $Y \subseteq [\![M]\!], |Y| \leq 2k$.*

*Proof.* For a $P_0$ matrix, we remark that its determinant is equivalent to the product of all singular values. For $Y \subseteq [\![M]\!]$ and $i \notin Y$, from the interlacing inequality of Eq. 50 we have that

$$F(Y \cup \{i\}) - F(Y) = \sum_{j=1}^{|Y|+1} \log\sigma_j' - \sum_{j=1}^{|Y|} \log\sigma_j \geq \log\sigma_{|Y|+1}' \geq \log\sigma_{\min} \tag{53}$$

where $\sigma_j'$ and $\sigma_j$ are the $j$-th largest singular value of $\boldsymbol{L}_{Y \cup \{i\}}$ and $\boldsymbol{L}_Y$, respectively. Similarly, using Eq. 51, we get

$$F(Y \cup \{i\}) - F(Y) \leq \log(\sigma_1'\sigma_2') - \log\sigma_{|Y|} \leq 2\log\sigma_{\max} - \log\sigma_{\min}. \tag{54}$$

$\square$

