# OpenReview forum: "Scalable Learning and MAP Inference for Nonsymmetric Determinantal Point Processes"
_ICLR.cc/2021/Conference — ICLR 2021 Oral_

### Official Review · AnonReviewer3 · 2020-10-27
**A good submission, it could be accepted once the author fix the bug in their experiments.**

**Rating:** 8
**Confidence:** 4

**Review:**

This paper propose a decomposition for non-symmetric determinantal point process (NDPP) kernels (M*M) which reduces the requirements of storage and running to linear in cardinality (M). Additionally, they derive a NDPP maximum a posteriori inference algorithm that applies to both their proposed kernel and the previous work (NDPP). In their experiments, they show both learning kernels and the MAP inference for subset selection on real-life datasets.

Pros:

	○ This paper is well-written and easy to follow.
	○ The author provide sufficient calculation process and relevant proofs of scalable NDPP.
	○ For the existing problems of traditional DPP method is time consuming, need large memory and could not apply to large set, the scalable NDPP really solves them (e.g., this method could run on Instacart and Million Song datasets). I think it is practical.


Major Concern:

	○ I have some doubts about the authenticity of the experimental results in Table 2, for the following reason: in the previous work, i.e., [1] Learning Nonsymmetric Determinantal Point Processes. Gartrell et al. Neurips2019. The results under average MPR have signifcant difference with same datasets and same hyperparameter settings. However, in NDPP paper,  for Amazon: Apparel, the MPR of sym dpp is 77.42±1.12, MPR of nonsym DPP is 80.32±0.75, for uk retail,  the MPR of sym dpp is 76.79±0.6, MPR of nonsym DPP is 79.45±0.57.  In this paper,   for Amazon: Apparel, the MPR of sym dpp is 62.63±1.81, MPR of nonsym DPP is 72.2±3.07, for uk retail,  the MPR of sym dpp is 69.95±1.32, MPR of nonsym DPP is 74.17±1.37. There is a huge gap btw these two versions. I would like to know what causes this gap.

---

> ### Author Response · Authors · 2020-11-17
> **Rebuttal feedback**
>
> Thank you for your feedback.  Regarding your questions about the experimental results in Table 2:
>
> - Performance gap between this paper and “Learning Nonsymmetric Determinantal Point Processes” (Gartrell et al., 2019):
>   - This reduction in MPR performance for the symmetric and nonsymmetric DPP baseline models is due to a fix for a bug in the computation of the MPR denominator; this bug was present in the original code for the NeurIPS 2019 paper.  The authors of (Gartrell et al., 2019) have fixed this bug in their public GitHub repository (as of Oct. 2020) and updated the arXiv version of their paper (https://arxiv.org/abs/1905.12962); the results in Table 1 in their current arXiv version match the results in our paper.

---

### Official Review · AnonReviewer4 · 2020-10-27
**Official Blind Review #4**

**Rating:** 7
**Confidence:** 4

**Review:**

Nonsymmetric determinantal point processes (NDPPs) received some attention recently because they allow modeling of both negative and positive correlations between items. This paper developed scalable learning and MAP inference algorithms with space and time complexity linear in ground set size, which is a huge improvement compared to previous approaches. Experimental results show that the algorithms scale significantly better, and can roughly match the predictive performance of prior work.

This is a well written paper and I recommend its acceptance. Scalable learning and MAP inference algorithms are important for the application of the NDPPs model, which seems promising compared with its symmetric counterpart in experiments.

I have some (minor) comments listed below.

1. In Lemma 1, the result is only proved for skew-symmetric matrices with even rank. Does it hold for odd rank matrices? This is important to support the claim that the new decomposition covers the $P_0^+$ space.

2. Equation (3) uses notation $\lambda_i$, which is already used in Lemma 1. This could cause confusion.

3. In the paragraph after Theorem 1, it is proposed to set $B$ = $V$ and relax $C$. Is this used in Section 4? If not, I would suggest moving it to the experiments section, and adding some comparison in Table 2 to show the impact of this simplification.

4. I cannot quite understand the last sentence before Lemma 2. What can be computed in $O(K^2)$ time?

5. The footnote in Table 1 might cause confusion because it can be mis-interpreted as a square.

6. In G.1, the first sentence after equation (13), do you mean when $M$ is odd or when $\ell$ is odd?

7. In equation (24), $X$ should be $B^TXB$

8. The inverse of $C$ appears in the gradient of $Z$. Is $C$ guaranteed to be invertible in the learning algorithm? And how are $V$, $B$, $C$ initialized in the algorithm?

9. In equation (31), please double check if we need the reciprocal in the denominator.

---

> ### Author Response · Authors · 2020-11-17
> **Rebuttal feedback**
>
> Thank you for the detailed comments.  Regarding your feedback:
>
> - Correctness of Lemma 1 with odd rank matrices:
>   - The rank of nonsingular skew-symmetric matrices is always even because all of its eigenvalues are purely imaginary and come in conjugate pairs. Hence, it is not necessary to consider odd rank matrices.  We have added text in the proof of Lemma 1 (Appendix G.1) to indicate this.
>
> - Eq. 3 uses notation $\lambda_i$:
>   - We have updated the text to use $\mu_i$ instead, to avoid confusion with $\lambda$ as used in Lemma 1.
>
> - Setting $B=V$ and relaxing $C$:
>   - We have moved the paragraph on this topic, which was originally found after Theorem 1 in Sec. 3, to the beginning of Sec. 5 (Experiments).
>
> - $O(K^2)$ time for computing Eq. 5:
>   - Using Lemma 2, the Eq. 5 can be computed in $O(K^2)$ time. The details are described in the first paragraph in page 5 and Algorithm 1.
>
> - Footnote in Table 1:
>   - We are now using a non-numeric character for the footnote character here, to avoid confusion.
>
> - In Appendix G.1, for the first sentence after Eq. 13:
>   - We mean when \ell is odd.  However, we have revised the proof for Lemma 1 (Appendix G.1), and have removed this remark.
>
> - Invertibility of $C$:
>   - The matrix $C$ is initialized from the standard Gaussian distribution, and thus it is not rank-deficient initially. We also observe empirically that it is always full-rank during optimization. To safely avoid singularity, we add a small perturbation to its diagonal, as $C + \epsilon I$, with $\epsilon = 10^{-5}$, if we detect that $C$ becomes singular.
>
> - Eq. 24: X should be B^T X B.  We have corrected this in the text.
>
> - Initialization of $V$, $B$, and $C$ in our NDPP learning algorithm:
>   - $C$ is initialized from the standard Gaussian distribution with mean 0 and variance 1.  $B$ (which we set equal to $V$) is initialized from the uniform(0,1) distribution.  We have updated the text in Sec. 5.2 to reflect this.
>
> - Eq. 31: The reciprocal is not needed in the denominator.  We have corrected this in the text.
>
> We have uploaded a revised version of our paper (with supplementary material included in the same PDF), which includes minor changes for a number of the points listed above.

---

> > ### Comment · AnonReviewer4 · 2020-11-22
> > **Response to rebuttal feedback**
> >
> > Thanks for the feedback and the revisions to the paper. I am satisfied to the response. Congratulations on the great work!

---

### Official Review · AnonReviewer2 · 2020-10-31
**A good submission with sound theoretical support and practical potential. I vote a strong acceptance.**

**Rating:** 9
**Confidence:** 5

**Review:**

This paper presented a novel kernel decomposition for nonsymmetric determinantal point processes, which enables linear time of inference and learning w.r.t. the cardinality of the ground set M. This is a significant improvement over previous arts and makes NDPP practical in relatively large datasets. The theoretical of the paper is solid and supportive to the main claim of the paper. This paper is well written and easy to follow. Even for readers without much theoretical background, this paper is still moderately friendly since the logic and insight are clear. I vote this paper a strong acceptance, except for some minor concerns as follows:

1) The authors employed a way to simplify the kernel decomposition below Theorem 1. However, it is unclear what the impact of this simplification is to the exactness of learning or inference. It would me a plus if the authors can give some theoretical analysis on the gap between such simplification and $P_0^+$.

2) I notice the authors provide both learning and inference procedures for NDPP. It seems these two procedures can formulate a way to learn latent NDPP in an Expectation-Maximization fashion. I would suggest the authors give some discussion about the feasibility of such integration, and this can be an interesting direction.

In general, I enjoy reading this paper and think this paper is insightful.

---

> ### Author Response · Authors · 2020-11-17
> **Rebuttal feedback**
>
> Thank you for the feedback and suggestions.  Regarding the points you raised:
> - Impact of $B = V$ simplification for our decomposition of the $L$ kernel:
>   - We are not aware of any theoretical approach to analyzing the impact of the $B = V$ restriction on the space of $P_0^+$ matrices that can be represented.  However, we empirically observe approximately the same predictive performance for the (Gartrell et al., 2019) NDPP and our approach with the $B=V$ restriction, as shown in Table 2 in our paper.  Since the (Gartrell et al., 2019) NDPP decomposition covers the full $P_0^+$ space, this empirical evidence suggests that our $B=V$ restriction does not significantly reduce coverage of the $P_0^+$ space.
>
> - Expectation-maximization (EM) for NDPP learning:
>   - We are not sure what AnonReviewer2 has in mind.  The only prior work that we are aware of regarding an EM algorithm for DPP kernel learning is https://arxiv.org/abs/1411.1088 (Gillenwater et al., 2014), which describes an EM algorithm for learning symmetric DPP kernels.  This method uses EM to optimize a variant of the DPP log-likelihood that is written in terms of the eigendecomposition of the kernel, where the intermediate hidden variable used in the log-likelihood is expressed in terms of the eigenvalues.  This approach assumes real eigenvalues and does not appear to be appropriate for NDPPs, since NDPP kernels may have complex eigenvalues.  So, it is not entirely clear to us how to formulate an EM algorithm for NDPP learning, nor how MAP inference can be applied to EM.  Can AnonReviewer2 please elaborate on their proposed EM approach?

---

> > ### Comment · AnonReviewer2 · 2020-11-20
> > **response to authors**
> >
> > I have read the response from the authors and still believe the paper is worth for a strong acceptance.
> > In general, I understand that finding the theoretical gap is very difficult and the corresponding workload can lead to another paper.
> > For the 2nd point, I was just curious if there's any possibility of extending the proposed framework to an EM fashion in terms of NDPP. Personally, I think finding an efficient optimization method for EM with NDPP can be an interesting topic.
> > Thank the authors for their response.

---

### Comment · Area_Chair1 · 2020-11-22
**Any last comments?**

Dear reviewers,

Thank you very much for your reviews.  The authors have given concrete responses to the concerns raised by reviewers.  Please acknowledge if you are satisfied with the response or speak up if you still have remaining concerns.

---

### Decision · Program_Chairs · 2021-01-07
**Final Decision**

**Decision:**

Accept (Oral)

**Comment:**

This paper proposes a technique of decomposing the nonsymmetric kernel of determinantal point processes, which enables inference and learning in time and space linear with respect to the size of the ground set.  This substantially improves upon existing work.  The proposed method is well supported both with theory and experiments.  All of the reviewers find that the contributions are significant, and no major flaws are identified through reviews and discussion.  The determinantal point process might not be one of the most popular topics in the ICLR community today but certainly is relevant.